# Improving LLM-Based Recommenders with Conservative Generative Flow Networks

Xuan Yu [1]  Feng Niu [1]  Rui Zhu [1]  Yudong Zhang [1,2]  Xu Wang [1,2]  Yang Wang [1,2]

## Abstract

Generative Flow Networks (GFlowNets) have recently been used to improve diversity and mitigate popularity bias in LLM-based recommender systems, yet most objectives are developed under online-style assumptions. In offline LLM-based recommendation, learning is constrained to a fixed logged dataset, yielding partial support over token transitions on the dataset-induced token-prefix DAG. Naively applying Sub-Trajectory Balance (SubTB) becomes non-identifiable and can arbitrarily allocate probability mass to unsupported regions. We formalize this failure and identify three sources of non-identifiability that induce distributional shift between the dataset-implied policy and the learned policy: (i) flow overestimation, (ii) forward mass leakage, and (iii) backward compensation. To address it, we propose CFlower, which introduces a conservative SubTB objective that explicitly penalizes unsupported forward flow mass, and combines it with dataset-constrained policy learning with on-policy sampling on the dataset-induced DAG for efficient training under offline constraints. Experiments on three Amazon recommendation datasets show that CFlower improves distributional matching and delivers a stronger accuracy–exposure trade-off than prior GFlowNet and SFT baselines, while serving as a more reliable reference policy for downstream RL fine-tuning.

## 1. Introduction

Recommender systems are a cornerstone of modern information platforms, shaping what users watch, read, and buy, and directly impacting user experience and business outcomes (Kohavi et al., 2020; Ricci et al., 2010). Classical collaborative filtering and learning-to-rank approaches provide effective foundations for personalization at scale (Koren et al., 2009; Rendle et al., 2012; He et al., 2017). With advances in sequential recommendation backbones (e.g., self-attentive and bidirectional Transformer models) (Kang & McAuley, 2018; Sun et al., 2019), *LLM-based* recommenders have emerged as a compelling paradigm: large language models can encode rich user histories and item semantics, and support next-item prediction via autoregressive generation or decoding-style ranking. These approaches have reported state-of-the-art performance (Bao et al., 2024; 2025; Gao et al., 2025; Jiang et al., 2024).

A common approach to training LLM-based recommenders is supervised fine-tuning (SFT), which maximizes the log-likelihood of logged next items given user histories (Bao et al., 2024; 2025). While SFT achieves strong accuracy, it primarily optimizes for user utility and can exacerbate popularity bias, leading to imbalanced exposure across items and reduced diversity (Jiang et al., 2024). This focus on imbalance is consistent with recent progress in imbalanced representation learning and dynamic balancing strategies (Wang et al., 2026; Wang et al.), and practical large-model adaptation also benefits from parameter-efficient designs such as token-wise low-rank projections (Li et al., 2025; 2026). To address these fairness and diversity concerns, recent works begin to apply Generative Flow Networks (GFlowNets) to LLM-based recommenders (Wang et al., 2025; Gao et al., 2025). GFlowNets provide a principled framework for learning generative policies that sample diverse, high-reward outcomes proportional to a reward function, naturally balancing accuracy with exposure fairness and diversity. By enforcing flow consistency over token-generation trajectories, GFlowNets guide the model toward more balanced item distributions while maintaining recommendation quality.

GFlowNet-based recommenders can be deployed in different learning settings, which fundamentally differ in how much interaction the agent is allowed (Sutton et al., 1998). In **online environments**, the agent interacts with the environment in real time, collecting new data and adapting its policy throughout training. However, deploying real online environments where algorithms interact with actual users is

---

[1]University of Science and Technology of China, Hefei, China [2]Suzhou Institute for Advanced Research, University of Science and Technology of China, Suzhou, China. Correspondence to: Xu Wang <wx309@ustc.edu.cn>, Yang Wang <angyan@ustc.edu.cn>.

*Proceedings of the $43^{rd}$ International Conference on Machine Learning*, Seoul, South Korea. PMLR 306, 2026. Copyright 2026 by the author(s).

costly and risky (Kohavi et al., 2020). Many online environments used for research are, in fact, *simulated* (Wang et al., 2025). They employ user models that mimic real behaviors and feedback, enabling iterative policy updates and exploration during training. For instance, GFlowGR (Wang et al., 2025) operates within a simulator, employing a collaborative model to estimate scores and using multiple reward signals derived from heuristic rules. Nonetheless, these simulated approaches remain susceptible to the *sim-to-real gap* (Tobin et al., 2017): a policy that performs well within the simulator may fail to generalize to real, dynamic user interactions and non-stationary environments. These gaps can stem from imperfect feedback modeling, covariate shifts, preference drift, or unmodeled business constraints.

By contrast, **offline environments** rely on static historical logs and strictly forbid online interaction or exploration during learning (Levine et al., 2020). Offline settings avoid the drawbacks of both simulated and real online environments, but bring their own challenges, including distributional shift (Levine et al., 2020) [1] and the inability to explore. According to Lin et al. (2023), most existing RL-based recommender systems rely on offline environments. However, despite the prevalence of offline scenarios, existing GFlowNet methods often reuse objectives developed under online-style assumptions, implicitly relying on exploratory interaction and broader transition support than what logged data can provide. This naive transfer can create mismatches between algorithmic assumptions and offline constraints, leading to distributional shift between the dataset-implied policy and the learned policy.

Motivated by this gap, we are, to our knowledge, the first to systematically formulate and address *offline GFlowNet learning in LLM-based recommenders*, establishing a principled framework tailored to offline recommender systems. We identify three factors that induce this distribution shift and propose a novel Conservative Sub-Trajectory Balance (CSubTB) objective to address them. Our contribution can be summarized as follows:

- We systematically formulate *offline GFlowNet learning in LLM-based recommenders* on a dataset-induced token-prefix DAG, and show that directly applying SubTB under partial support leads to objective-level non-identifiability and distributional shift. Our analysis identifies three fundamental mechanisms: *flow*

---

[1] We use *distributional shift* in a GFlowNet-specific sense: a mismatch between the data-implied (support-constrained) token-transition structure/policy and the learned generative policy/flows, often caused by probability/flow mass leaking to unsupported actions in the dataset-induced prefix DAG. This differs from the standard offline RL notion, where "distribution shift" typically refers to the covariate shift between the behavior-policy data distribution and the state–action distribution induced by the learned policy.

*overestimation*, *forward mass leakage*, and *backward compensation*.

- We propose *Conservative Sub-Trajectory Balance (CSubTB)*, which explicitly penalizes unsupported forward flow mass to suppress spurious probability leakage and mitigate the offline non-identifiability issues.

- We combine CSubTB with dataset-constrained training and on-policy sampling on the dataset-induced DAG, improving learning efficiency while respecting offline constraints.

- Experiments on three real-world Amazon recommendation datasets demonstrate that CFlower achieves better distributional matching and more balanced exposure while maintaining strong recommendation accuracy.

## 2. Preliminaries

### 2.1. LLM-Based Recommendation

Let $\mathcal{D} = \{(u_k, \mathbf{h}_k, i_k, r_k)\}_{k=1}^N$ denote a logged dataset of user–item interactions, where $u_k$ is a user, $\mathbf{h}_k = [(i_{k,1}, r_{k,1}), \cdots, (i_{k,|\mathbf{h}_k|}, r_{k,|\mathbf{h}_k|})]$ represents the historical interaction sequence of user $u_k$, $i_k$ is the target item, and $r_k$ is the corresponding interaction signal (e.g., rating or implicit feedback). Let $\mathcal{I}$ denote the set of all unique items appearing in $\mathcal{D}$. Following recent LLM-based recommendation approaches, each item $i \in \mathcal{I}$ is represented as a sequence of tokens $i = [i^1, i^2, \ldots, i^L]$, where each token $i^l$ belongs to the LLM vocabulary $\mathcal{V}$. Recommendation is thus formulated as an autoregressive sequence generation problem, where the model generates item tokens sequentially.

Given an autoregressive LLM parameterized by $\theta$, the conditional likelihood of an item is

$$p_\theta(i \mid c) = \prod_{t=1}^L p_\theta(i^t \mid c, i^{<t}). \tag{1}$$

### 2.2. A GFlowNet Perspective on Recommendation

Generative Flow Networks (GFlowNets) define a generative process over a directed acyclic graph (DAG) $\mathcal{G} = (\mathcal{S}, \mathcal{A})$ with an initial state $s_0$ and a set of terminal states $\mathcal{X} \subset \mathcal{S}$. A *trajectory* $\tau = (s_0 \rightarrow s_1 \rightarrow \cdots \rightarrow s_T = x)$ is generated by a forward policy $P_F(s_{t+1} \mid s_t)$, together with a backward policy $P_B(s_t \mid s_{t+1})$. Each terminal state $x \in \mathcal{X}$ is assigned a non-negative reward $R(x)$, and the learning objective is to train $P_F$ such that the induced terminal-state distribution satisfies

$$P_F(x) = \sum_{\tau:x \in \tau} \prod_{t=0}^{|\tau|} P_F(s_{t+1}|s_t) \ \propto \ R(x). \tag{2}$$

In the context of LLM-based recommendation, autoregressive item generation induces a natural GFlowNet state space. Each state $s \in \mathcal{S}$ corresponds to a token-prefix sequence, and an action $a = (s, s') \in \mathcal{A}$ represents an *add-a-token* transition, where $s' = s \oplus v$ for a token $v \in \mathcal{V}$. The empty prefix corresponds to the initial state, and terminal states correspond to complete item sequences.

Under offline recommendation, supervision is available only for item sequences observed in the logged dataset. Consequently, the dataset induces a subgraph $\mathcal{G}_D = (\mathcal{S}_D, \mathcal{A}_D)$ of the full prefix DAG, where $\mathcal{A}_D(s)$ denotes the set of tokens observed to follow prefix $s$ in $\mathcal{D}$. Transitions outside $\mathcal{A}_D$ are not directly constrained by data.

Many GFlowNet training objectives are derived from flow consistency described in Eq. 2. In this work, we focus on the Subtrajectory Balance (SubTB) objective (Madan et al., 2023). In GFlowNets, the flow of terminal states is represented as the corresponding reward. For an intermediate state $s$, we have $F(s) = \sum_{s' \in child(s)} F(s')$. We define $f(s) = \log F(s)$. For any sub-trajectory $\tau_{i:j} = (s_i, \ldots, s_j)$, SubTB enforces

$$f(s_i) + \sum_{t=i}^{j-1} \log P_F(s_{t+1} \mid s_t) = f(s_j) + \sum_{t=i}^{j-1} \log P_B(s_t \mid s_{t+1}), \quad (3)$$

A typical SubTB loss aggregates the squared residuals between $f(s_i)$ and $f(s_j)$ in Eq. 3 over sampled subtrajectories. Concretely, for a sampled trajectory $\tau = (s_0, \ldots, s_T)$, a common SubTB loss is

$$\mathcal{L}_{\text{SubTB}}(\tau) = \sum_{0 \le i < j \le T} \lambda^{j-i} \Delta_{i:j}^2,$$
$$\Delta_{i:j} = f(s_i) - f(s_j) \quad (4)$$
$$+ \sum_{t=i}^{j-1} \Big( \log P_F(s_{t+1} \mid s_t) - \log P_B(s_t \mid s_{t+1}) \Big).$$

where $\lambda > 0$ controls the relative emphasis on longer vs. shorter sub-trajectories.

## 3. Methodology

### 3.1. Failure of SubTB in Offline Dataset-induced DAGs

We analyze the behavior of the Subtrajectory Balance (SubTB) objective when applied to offline LLM-based recommendation under the GFlowNet formulation introduced before. Our analysis focuses on the dataset-induced DAG $\mathcal{G}_D = (\mathcal{S}_D, \mathcal{A}_D)$, where only a subset of prefix transitions (i.e., token transitions) is observed in logged data.

We consider the standard GFlowNet parameterization in which a shared neural network (e.g., an LLM) defines the forward policy $P_F(a \mid s)$ over all tokens $a \in \mathcal{V}$ at each prefix state $s$. The backward policy $P_B$ is assumed to be independent of the forward parameters, as is common in

prefix-based generation settings. This setting isolates the effect of the SubTB objective itself, without introducing additional degrees of freedom through backward parameterization.

Importantly, while the forward policy assigns probabilities to all tokens in the vocabulary, SubTB constraints are only enforced on transitions and subtrajectories that are observed in the dataset. Transitions outside $\mathcal{A}_D$ do not appear in the training samples. We analyze the structure of the SubTB constraints induced by a collection of subtrajectories $\mathcal{C}$ on the prefix DAG. These constraints form a system of equations over the state flows $\{F(s)\}$ and forward transition probabilities $\{P_F(a \mid s)\}$. We show that this system admits intrinsic degrees of freedom due to three independent invariance mechanisms.

**Proposition 1** (State-Flow Scaling Ambiguity). *Let $(P_F, F)$ satisfy all SubTB constraints in $\mathcal{C}$. Consider the graph $\mathcal{H}_\mathcal{C}$ whose nodes are states and whose edges connect states that co-occur in at least one constrained subtrajectory in $\mathcal{C}$. Let $\{\mathcal{K}_m\}_{m=1}^M$ be the connected components of $\mathcal{H}_\mathcal{C}$. For any function $\alpha : \mathcal{S} \to \mathbb{R}_{>0}$ that is constant on each $\mathcal{K}_m$ and satisfies $\alpha(s_0) = 1$, define $\tilde{F}(s) = \alpha(s)F(s)$ and $\tilde{P}_F = P_F$. Then $(\tilde{P}_F, \tilde{F})$ also satisfies all SubTB constraints in $\mathcal{C}$.*

Proposition 1 shows that SubTB does not uniquely determine the relative scaling of flows across constraint-disconnected regions of the state space.

**Proposition 2** (Local Transition Redistribution). *Let $(P_F, F)$ satisfy all SubTB constraints in $\mathcal{C}$. Fix a state $s$ and let $\mathcal{A}_\mathcal{C}(s)$ denote the set of outgoing transitions from $s$ that appear in at least one constrained subtrajectory in $\mathcal{C}$. Assume $\mathcal{A}_\mathcal{C}(s)$ is a strict subset of all outgoing transitions from $s$. Then there exists a one-parameter family of modified pairs $(P_F^{(\kappa)}, F^{(\kappa)})$, with $\kappa \in (0, 1)$, such that all SubTB constraints in $\mathcal{C}$ remain satisfied while the forward transition distribution at $s$ changes.*

Proposition 2 implies that SubTB does not uniquely fix the allocation of outgoing probability mass at individual states when some transitions do not participate in any constraint.

**Proposition 3** (Backward Compensation Invariance). *Let $(P_F, F)$ satisfy SubTB under a backward policy $P_B$. For any function $g : \mathcal{S} \to \mathbb{R}$ with $g(s_0) = 0$, define transformed quantities*

$$f'(s) = f(s) + g(s), \quad (5)$$
$$\log P'_F(s' \mid s) = \log P_F(s' \mid s) + g(s) - g(s'), \quad (6)$$
$$\log P'_B(s \mid s') = \log P_B(s \mid s') + g(s') - g(s). \quad (7)$$

*Then $(P'_F, F')$ together with $P'_B$ satisfies exactly the same SubTB constraints as $(P_F, F, P_B)$.*

Proposition 3 shows that SubTB admits a gauge-like symmetry in which forward, backward, and flow variables can

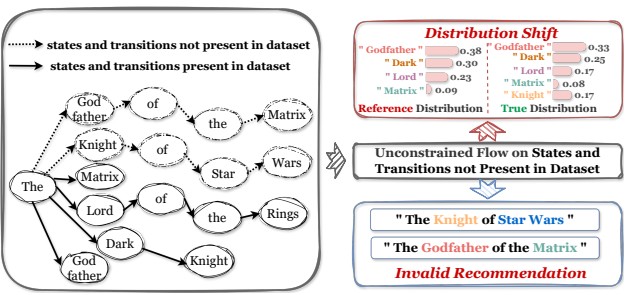

*Figure 1.* Illustration of a failure mode in training GFlowNets with LLM-based policies. During training, logits corresponding to unsupported transitions may drift due to their semantic similarity to supported transitions. As a result, probability mass is unintentionally assigned to out-of-support actions, leading to unexpected distributional shift.

be jointly transformed without affecting the objective.

The three propositions above characterize independent invariances of the SubTB constraint system. Together, they imply that the solution set of SubTB is non-identifiable.

**Theorem 1** (Non-identifiability of SubTB). *Assume the dataset-induced DAG contains at least one non-terminal state and multiple distinct trajectories from the initial state to terminal states. Then the Subtrajectory Balance objective admits a non-trivial equivalence class of solutions: there exist infinitely many $(P_F, F)$ pairs achieving identical SubTB loss values while inducing different trajectory and terminal-state distributions.*

Theorem 1 establishes that minimizing SubTB does not uniquely determine the forward policy, even under perfect optimization. The ambiguity arises from intrinsic degrees of freedom in the flow and transition variables that are not fixed by SubTB constraints. Proofs for the propositions and theorem can be found at Appendix C.

As shown in Figure 1, when using LLM-based policies as the forward policy of a GFlowNet, non-identifiability is further exacerbated because semantically similar items can induce nearly indistinguishable trajectory likelihoods, allowing probability mass to drift toward unsupported transitions without being reliably penalized.

### 3.2. Conservative SubTB Formulation

To address these issues, we introduce a *Conservative SubTB* objective that explicitly suppresses forward *flow mass* allocated to transitions unsupported by the offline dataset. Let $\mathcal{A}_{\mathcal{D}}(s)$ denote the set of actions observed in the logged dataset $\mathcal{D}$ at state $s$. For any pair $(s, a)$ with $a \notin \mathcal{A}_{\mathcal{D}}(s)$, we call the transition *unsupported* and treat the associated forward flow mass $F(s)P_F(a \mid s)$ as spurious. Accordingly,

the total unsupported outgoing flow from state $s$ is

$$F(s) \sum_{a \notin \mathcal{A}_{\mathcal{D}}(s)} P_F(a \mid s), \tag{8}$$

which measures how much flow is "leaked" to actions without empirical support. In general, Proposition 3 suggests that unsupported forward transitions should be accompanied by corresponding penalties on the backward policy. However, in our setting, the token-level decision process induces a tree-structured state space, under which each state admits a unique parent. As a result, all backward transitions are necessarily supported, and no additional regularization on the backward policy is required.

For a partial trajectory $\tau_{i:j} = (s_i, s_{i+1}, \ldots, s_j)$, we penalize the unsupported flows for each state within the sub-trajectory. The Conservative SubTB loss for sub-trajectory $\tau_{i:j}$ is:

$$\mathcal{L}_{\text{C-SubTB}}(\tau_{i:j}) = \mathcal{L}_{\text{SubTB}}(\tau_{i:j})$$
$$+ \alpha \lambda^{j-i} \sum_{s \in \tau_{i:j}} \sum_{a \notin \mathcal{A}_{\mathcal{D}}(s)} \phi(F(s)P_F(a \mid s)), \tag{9}$$

where $\alpha > 0$ controls the overall strength of the conservative regularizer, $\lambda > 0$ optionally reweights the penalty by sub-trajectory length via $\lambda^{j-i}$ (with $\lambda > 1$ emphasizing longer sub-trajectories and $\lambda < 1$ emphasizing shorter ones), and $\phi(\cdot)$ is a non-negative increasing function (e.g., $\phi(p) = p$ or $\phi(p) = p^2$) that penalizes unsupported forward flow mass.

For a complete trajectory $\tau = (s_0, s_1, \ldots, s_T)$ from the dataset, SubTB enumerates all sub-trajectory pairs $(s_i, s_j)$ where $0 \leq i < j \leq T$, and enforces flow balance for each sub-trajectory. The Conservative SubTB loss for a complete trajectory $\tau$ is then:

$$\mathcal{L}_{\text{C-SubTB}}(\tau) = \sum_{0 \leq i < j \leq T} \mathcal{L}_{\text{C-SubTB}}(\tau_{i:j}). \tag{10}$$

This regularization directly targets the offline non-identifiability: it shrinks the null space of SubTB by assigning an explicit cost to any flow that cannot be justified by $\mathcal{D}$. Unlike pure path-level balance constraints, the conservative term makes the optimization *support-aware*, discouraging probability mass leakage to unsupported actions that would otherwise be unconstrained. From a flow-matching perspective, it enforces a conservative decomposition in which only flow mass that can be matched to empirical evidence is retained, while unsupported flow is systematically driven toward zero.

### 3.3. Constrained On-policy Learning

Typically, in offline learning, trajectories are sampled directly from the logged dataset $\mathcal{D}$ and fed into the model for

training. In COFlownet (Zhang et al., 2025), the default approach adopts off-policy learning with uniform sampling from the dataset. However, this uniform sampling strategy leads to low efficiency, as it treats all trajectories equally regardless of their quality or relevance. A more reasonable approach is to sample trajectories proportional to their rewards, which naturally prioritizes high-reward trajectories and improves training efficiency.

We propose constrained on-policy learning, where trajectories are sampled from the dataset-induced DAG by using the forward policy $P_F$ to guide token generation, rather than directly replaying trajectories from $\mathcal{D}$. We define a constrained forward policy

$$\tilde{P}_F(a \mid s) \propto \begin{cases} P_F(a \mid s), & a \in \mathcal{A}_\mathcal{D}(s), \\ 0, & \text{otherwise}, \end{cases} \quad (11)$$

with renormalization over $\mathcal{A}_\mathcal{D}(s)$. Specifically, at each step, $\tilde{P}_F$ selects a next-token (add-a-token) action conditioned on the current token prefix, generating trajectories (complete item titles). These dynamically generated token sequences are then used for model training. On-policy learning is more efficient than off-policy learning because $P_F$ actively explores high-value regions of the token-prefix space, generating samples that are more likely to reduce the objective loss and accelerating convergence compared to passive uniform sampling from the dataset. In our experiments, we compare the performance of both learning settings.

# 4. Experiments

In this section, we aim to address the following research questions:

**RQ1**: How well does CFlower match the target distribution compared with non-offline methods?

**RQ2**: How does CFlower perform across different evaluation metrics?

**RQ3**: Does CFlower exhibit improved generalizability when used as a reference policy for reinforcement learning methods such as SDPO?

**RQ4**: How do different components of our model affect recommendation performance?

**RQ5**: How robust is the model, and to what extent do the conservative terms affect performance under varying hyperparameter settings?

To answer these questions, we evaluate CFlower on three real-world datasets, compare it against multiple baselines, and conduct ablation studies to assess the effectiveness and robustness of the proposed modules.

## 4.1. Experimental Setups

### 4.1.1. DATASETS

We evaluate CFlower on three real-world datasets: CDs and Vinyl, Video Games, and Movies and TV. These datasets are derived from the public Amazon product review corpus (McAuley et al., 2015) and correspond to three representative product domains with different item semantics and interaction patterns. Each dataset contains timestamped user-item interactions (e.g., reviews/purchases) with associated ratings. Following the preprocessing protocol in Flower (Gao et al., 2025), we truncate interactions by time, filter out users and items with fewer than five interactions, cap the maximum sequence length at 10, and chronologically split each dataset into training/validation/test sets with an 8:1:1 ratio.

### 4.1.2. BASELINES

We compare CFlower against a representative traditional sequential recommender, several supervised fine-tuning (SFT) LLM recommenders, and one GFlowNet-based method. We follow the official implementations and tune hyperparameters on the validation set under the same data split and candidate construction. **SASRec** (Kang & McAuley, 2018): a Transformer-based sequential recommender that predicts the next item from a user's interaction history. **BIGRec** (Bao et al., 2025): an SFT-based LLM recommender that generates the next item conditioned on a textualized user history. **Temp** (Bao et al., 2024): a template-based SFT/prompting baseline that formats user histories with fixed instruction templates. $D^3$ (Bao et al., 2024): an SFT baseline that directly learns recommendation decisions from logged interactions. **IFairLRS** (Jiang et al., 2024): a fairness-aware LLM recommender trained with additional fairness-oriented objectives or constraints. **Flower** (Gao et al., 2025): a flow-guided tuning approach that leverages process supervision for LLM-based recommendation.

### 4.1.3. EVALUATION METRICS

We evaluate both recommendation accuracy and distributional quality. For accuracy, we report NDCG@$K$ and HR@$K$ on the held-out test set. Given the ranked list $\pi_{1:K}$ and the ground-truth next item $y$, we compute DCG@$K = \sum_{i=1}^{K} \frac{\mathbb{I}[\pi_i = y]}{\log_2(i+1)}$ and NDCG@$K$ = DCG@$K$/IDCG@$K$. HR@$K$ is defined as $\mathbb{I}[y \in \pi_{1:K}]$.

Beyond accuracy, we assess how recommendations distribute utility across item groups. Following common practice, we split items into popularity-based groups (e.g., head vs. tail) according to training-set interaction counts. For a group $g$, we compute its discounted group utility as $\text{GU}_g@K = \sum_{i=1}^{K} \frac{\mathbb{I}[\pi_i \in g]}{\log_2(i+1)}$, aggregated over all users' top-$K$ lists. We then quantify *utility imbalance* via DGU@$K =$

$\frac{1}{|G|} \sum_{g \in G} \left| \mathrm{GU}_g@K - \overline{\mathrm{GU}}@K \right|$ (mean deviation) and $\mathrm{MGU}@K = \max_{g \in G} \left| \mathrm{GU}_g@K - \overline{\mathrm{GU}}@K \right|$ (maximum deviation), where $\overline{\mathrm{GU}}@K = \frac{1}{|G|} \sum_{g \in G} \mathrm{GU}_g@K$. Thus, *smaller* DGU/MGU indicates a more balanced allocation of utility across groups. To measure balance of exposure, we also compute the entropy $H = -\sum_{g \in G} p_g \log p_g$, where $p_g$ is the proportion of recommended items (over all users and top-$K$ lists) that fall into group $g$. Finally, we report TTR@$K$, the fraction of recommended items in the tail group among the top-$K$ positions. Higher $H$ and TTR indicate more balanced exposure and better long-tail coverage.

## 4.2. Distributional Quality (RQ1).

In RQ1, we evaluate whether a method can match a *global* target distribution over items, without conditioning on user histories (see Appendix for the history-based setups in RQ2–RQ5). Concretely, we define the target as the empirical item-frequency distribution in the logged data: $p_{\text{target}}(i) \propto \sum_{(u,\mathbf{h},y,r) \in \mathcal{D}} \mathbb{I}[y = i]$, where $y$ is the logged next item. For each method, we obtain an induced marginal distribution $p_{\text{model}}(i)$ by autoregressively generating item titles under the same item catalog, then mapping each complete generation to an item ID (and treating unmatched generations as uncovered). We define each distribution's *support* as the set of items with non-zero probability mass, and report both support overlap and probability matching on the intersection (common-item support), after restricting and re-normalizing. We evaluate how well each method matches the target next-item distribution by comparing their probabilities on the *common-item* support (i.e., items shared by the model and the target). Fig. 2 summarizes the discrepancy via $\log_{10}\left(p_{\text{model}}/p_{\text{target}}\right)$: mass concentrated around 0 indicates close agreement, whereas heavy tails reveal systematic over- or under-allocation. Across method–group pairings, CFlower exhibits a visibly tighter concentration near 0 and reduced tail mass than BIGREC, PPO, and Flower, suggesting better calibration on common-item transitions.

Table 1 highlights two complementary aspects: *support overlap* and *probability matching on the overlap*. CFlower achieves the strongest support overlap with the target, attaining the largest number of common items (1229) and the highest Jaccard similarity (0.854). By contrast, DPO attains smaller distances once restricted to the common-item overlap (e.g., Jensen–Shannon divergence 0.073 and TV distance 0.226), but it shows the weakest overlap overall (639 common items; Jaccard 0.444), suggesting limited coverage of the target support. Compared with BIGREC/PPO/Flower, CFlower improves distributional matching while preserving overlap, e.g., reducing TV/L1 distances to 0.332/0.664 and achieving a high within-5× mass of 0.919 with correlation 0.775. Overall, CFlower offers a favorable trade-off: strong support alignment with the target and competitive distributional closeness on the common-item region.

*Table 1.* Distributional matching between each method and the target next-item distribution on the *common-item* support. The target $Q = p_{\text{target}}$ is the empirical (unconditional) item-frequency distribution in the logged data, and $P = p_{\text{model}}$ is the induced marginal distribution from each method's generations. We restrict both distributions to common items and re-normalize. We report support overlap (No. common items; Jaccard) and probability-matching distances (JS, KL, TV, $\ell_1$), as well as log-ratio summaries where $r = \log_{10}(P/Q)$: median/mean/95th percentile of $|r|$, and Within-$\alpha\times$ mass = $\Pr(|r| \leq \log_{10} \alpha)$. Best/second-best marks exclude methods with Jaccard similarity $< 0.5$.

| Metric | BIGREC | PPO | DPO | Flower | CFlower |
|---|---|---|---|---|---|
| No. common items (↑) | 745 | 832 | 639 | 1150 | **1229** |
| Jaccard similarity (↑) | 0.518 | 0.578 | 0.444 | 0.799 | **0.854** |
| Jensen–Shannon divergence (↓) | 0.190 | 0.384 | 0.073 | 0.231 | **0.144** |
| KL($P\|Q$) (↓) | 0.879 | 2.089 | 0.294 | 1.402 | **0.862** |
| KL($Q\|P$) (↓) | 1.047 | 1.697 | 0.354 | 0.993 | **0.615** |
| Total variation (TV) distance (↓) | 0.420 | 0.636 | 0.226 | 0.436 | **0.332** |
| $\ell_1$ distance (↓) | 0.840 | 1.271 | 0.452 | 0.872 | **0.664** |
| Median $|r|$ (↓) | 0.324 | 0.574 | 0.223 | 0.270 | **0.247** |
| Mean $|r|$ (↓) | 0.406 | 0.583 | 0.309 | 0.357 | **0.310** |
| 95th percentile of $|r|$ (↓) | 1.036 | 1.204 | 0.891 | 0.985 | **0.808** |
| Within 1.25× mass (↑) | 0.148 | 0.072 | 0.252 | 0.176 | **0.202** |
| Within 2× mass (↑) | 0.472 | 0.244 | 0.613 | 0.549 | **0.591** |
| Within 5× mass (↑) | 0.830 | 0.643 | 0.892 | 0.874 | **0.919** |
| Within 10× mass (↑) | 0.944 | 0.880 | 0.973 | 0.954 | **0.974** |
| Pearson correlation (↑) | 0.670 | 0.656 | 0.801 | 0.671 | **0.775** |

## 4.3. Next-item recommendation performance(RQ2).

We evaluate the proposed method on next-item recommendation across three datasets during the supervised fine-tuning (SFT) stage. Table 2 reports both ranking accuracy (NDCG@5 and HR@5) and distributional quality metrics (DGU/MGU, exposure entropy $H$, and tail coverage TTR). Overall, CFlower achieves strong accuracy while maintaining competitive distributional quality, with small standard deviations indicating stable performance.

On **Video Games** and **Movies and TV**, CFlower attains the best accuracy among all methods (NDCG/HR 0.0646/0.0841 and 0.1036/0.1254, respectively). Compared with Flower, CFlower improves NDCG from 0.0546 to 0.0646 on Video Games and from 0.0956 to 0.1036 on Movies and TV, while also improving HR ($0.0803 \rightarrow 0.0841$; $0.1191 \rightarrow 0.1254$). In terms of distributional quality, CFlower remains competitive on entropy and tail exposure (e.g., $H = 7.82$, TTR $= 0.006$ on Video Games; $H = 9.028$, TTR $= 0.023$ on Movies and TV), suggesting that the accuracy gains do not come from collapsing exposure to a narrow set of popular items.

On **CDs and Vinyl**, CFlower remains competitive, improving over Flower on both NDCG and HR (0.0737/0.0923 vs. 0.0702/0.0891) but trailing the strongest accuracy-oriented baseline ($D^3$). We attribute this gap to the different optimization targets: $D^3$ is designed to directly maximize next-item ranking accuracy on the logged supervision, which can be particularly advantageous when the next-item distribution is highly concentrated and easier to exploit with accuracy-only training. However, $D^3$ yields larger DGU/MGU values

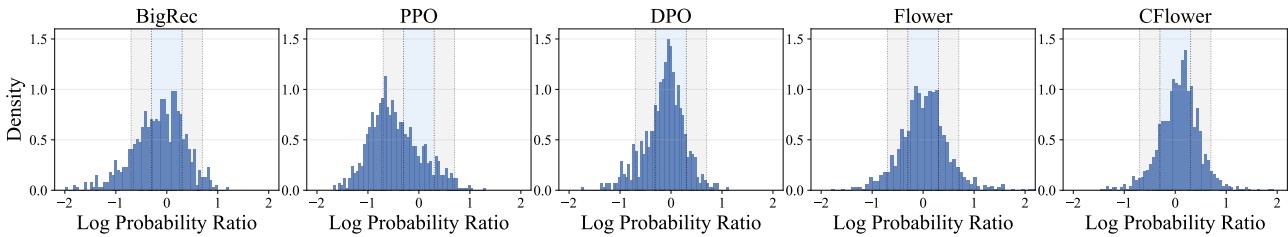

*Figure 2.* **Model–target distribution gap, visualized as log-ratio histograms on common items.** Each subplot shows the histogram of $\log_{10}\left(p_{\text{model}}/p_{\text{target}}\right)$ for a method–group pair, computed on *common items* that appear in both the model support and the target support. Positive values indicate over-allocation ($p_{\text{model}} > p_{\text{target}}$), while negative values indicate under-allocation. Shaded regions mark the fraction of probability mass whose ratio stays within $2\times$ or $5\times$ of the target; tighter concentration around 0 indicates better distributional matching.

*Table 2.* Recommendation quality (RQ2) on three datasets. We report the mean and standard deviation of 5 different random seeds. Best results are in bold and second-best results are underlined.

| Method | CDs and Vinyl NDCG(↑) | HR(↑) | DGU(↓) | MGU(↓) | H(↑) | TTR(↑) | Video Games NDCG(↑) | HR(↑) | DGU(↓) | MGU(↓) | H(↑) | TTR(↑) | Movies and TV NDCG(↑) | HR(↑) | DGU(↓) | MGU(↓) | H(↑) | TTR(↑) |
|---|---|---|---|---|---|---|---|---|---|---|---|---|---|---|---|---|---|---|
| SASRec | 0.0642±.0015 | 0.0847±.0009 | 0.182±.003 | 0.038±.001 | 9.167±.227 | 0.122±.003 | 0.0371±.0010 | 0.0552±.0006 | 0.169±.003 | 0.033±.000 | 8.231±.118 | 0.051±.001 | 0.0899±.0009 | 0.1066±.0015 | 0.139±.003 | 0.034±.001 | 8.887±.128 | 0.165±.004 |
| BIGRec | 0.0574±.0015 | 0.0717±.0007 | 0.216±.006 | 0.044±.001 | 5.907±.103 | 0.006±.000 | 0.0328±.0010 | 0.0472±.0008 | 0.153±.002 | 0.030±.000 | 7.513±.202 | 0.004±.000 | 0.0927±.0006 | 0.1131±.0010 | 0.127±.003 | 0.029±.001 | 8.285±.146 | 0.018±.000 |
| Temp | 0.0497±.0013 | 0.0615±.0014 | 0.225±.006 | 0.046±.001 | 6.209±.149 | 0.006±.000 | 0.0309±.0004 | 0.0447±.0007 | 0.130±.001 | 0.026±.000 | 7.341±.088 | 0.004±.000 | 0.0849±.0009 | 0.1058±.0008 | 0.141±.002 | 0.027±.000 | 8.113±.125 | 0.018±.001 |
| D3 | **0.0810**±.0019 | **0.0996**±.0022 | 0.353±.005 | 0.072±.002 | 7.641±.101 | 0.013±.000 | 0.0411±.0005 | 0.0602±.0014 | 0.219±.004 | 0.039±.001 | 7.575±.205 | 0.005±.000 | 0.1002±.0015 | 0.1218±.0013 | 0.148±.002 | 0.034±.001 | 8.336±.119 | 0.020±.000 |
| Flower | 0.0702±.0006 | 0.0891±.0009 | 0.076±.002 | 0.021±.000 | 7.928±.029 | 0.013±.000 | 0.0546±.0011 | 0.0803±.0016 | 0.108±.000 | 0.023±.001 | 7.763±.080 | 0.005±.000 | 0.0956±.0013 | 0.1191±.0025 | 0.075±.001 | 0.026±.001 | 8.799±.108 | 0.023±.000 |
| CFlower | 0.0737±.0014 | 0.0923±.0027 | **0.071**±.001 | **0.019**±.000 | 8.132±.163 | 0.015±.000 | **0.0646**±.0018 | **0.0841**±.0023 | 0.106±.002 | 0.022±.001 | 7.820±.070 | 0.006±.000 | **0.1036**±.0016 | **0.1254**±.0026 | 0.063±.002 | 0.025±.001 | 9.028±.090 | 0.024±.000 |

across datasets, indicating *more imbalanced* group utility, whereas CFlower tends to achieve smaller DGU/MGU along with higher exposure entropy $H$ and better tail coverage than Flower (e.g., $H = 8.132$ and TTR $= 0.015$ vs. 7.928 and 0.013 on CDs and Vinyl). Overall, these results highlight a favorable accuracy–exposure trade-off delivered by CFlower across diverse domains.

In particular, compared with Flower, CFlower consistently improves NDCG/HR while maintaining higher exposure diversity ($H$) and long-tail coverage (TTR), indicating a strictly better overall balance.

### 4.4. Performance as a Reference Policy (RQ3)

Many offline RL-style recommenders rely on a *reference policy* to stabilize optimization and prevent degenerate distribution shifts (e.g., via KL regularization or preference-based objectives). Since CFlower is trained to better respect offline support and match the target distribution (RQ1), we ask whether it can serve as a *stronger reference* for downstream RL fine-tuning, thereby improving both accuracy and exposure quality.

We instantiate several representative RL objectives (PPO, S-DPO, RosePO, and DMPO) and vary only the reference policy. We denote by B_X, F_X, and C_X the same RL algorithm $X$ using BIGREC, Flower, and CFlower as the reference policy, respectively. All models are evaluated with the same metrics as in RQ2.

Table 3 shows that using CFlower as the reference policy consistently improves downstream RL performance, especially for preference-based objectives. For example, with S-DPO, the CFlower-referenced variant achieves the best

ranking accuracy on all three datasets (e.g., NDCG/HR 0.0805/0.0965 on CDs and Vinyl, 0.0737/0.0914 on Video Games, and 0.1068/0.1265 on Movies and TV), outperforming both BIGREC- and Flower-referenced counterparts. Notably, these gains do not come at the cost of exposure collapse: the CFlower-referenced RL models typically maintain competitive or higher entropy $H$ and tail coverage (TTR), indicating a better accuracy–diversity trade-off after RL fine-tuning. Overall, the results support that CFlower provides a more reliable and generalizable reference policy, which translates into more effective and stable RL optimization in offline recommendation.

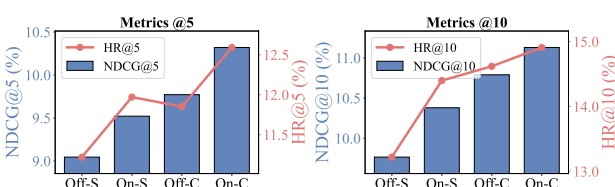

*Figure 3.* Impact of offline constraints and on-policy learning on recommendation performance. "Off" denotes off-policy learning and "On" denotes on-policy learning; "S" represents using SubTB objective and "C" represents using CSubTB objective.

### 4.5. Ablation of Offline Constraints and On-Policy Learning (RQ4)

We ablate offline constraints and on-policy learning in Fig. 3 by comparing off-policy vs. on-policy training under SubTB (S) and conservative SubTB (C). Removing offline constraints performs worst, showing that restricting optimization to data-supported transitions is critical under distribution shift. With constraints enabled, both variants im-

*Table 3.* RL Fine-tuning Performance with Different Reference Policies (RQ3) (mean ± std). B=BIGRec, F=Flower, C=CFlower. We report the mean and standard deviation of 5 different random seeds. Best results are in bold and second-best results are underlined.

| Method | CDs and Vinyl | | | | | | Video Games | | | | | | Movies and TV | | | | | |
|---|---|---|---|---|---|---|---|---|---|---|---|---|---|---|---|---|---|---|
| | NDCG(↑) | HR(↑) | DGU(↓) | MGU(↓) | H(↑) | TTR(↑) | NDCG(↑) | HR(↑) | DGU(↓) | MGU(↓) | H(↑) | TTR(↑) | NDCG(↑) | HR(↑) | DGU(↓) | MGU(↓) | H(↑) | TTR(↑) |
| B_PPO | 0.0521±.0006 | 0.0637±.0005 | 0.248±.004 | 0.050±.001 | 5.677±.14 | 0.005±.000 | 0.0279±.0008 | 0.0398±.0005 | 0.188±.004 | 0.035±.000 | 7.199±.104 | 0.004±.000 | 0.0874±.0009 | 0.1077±.0015 | 0.178±.004 | 0.034±.001 | 8.125±.117 | 0.016±.000 |
| B_S-DPO | 0.0716±.0019 | 0.0913±.0009 | 0.103±.003 | 0.025±.001 | 8.582±.143 | 0.016±.000 | 0.0675±.0020 | 0.0902±.0004 | 0.083±.001 | 0.020±.00 | 8.264±.222 | 0.008±.000 | 0.1043±.0027 | 0.1239±.0030 | **0.070±.001** | 0.022±.001 | 9.113±.159 | 0.025±.001 |
| B_RosePO | 0.0638±.0017 | 0.0806±.002 | 0.106±.03 | 0.023±.001 | 8.585±.208 | 0.017±.000 | 0.0594±.0009 | 0.0780±.0012 | 0.288±.003 | 0.057±.001 | 8.504±.103 | 0.008±.000 | 0.1004±.0023 | 0.1170±.0020 | 0.146±.003 | 0.030±.00 | 9.301±.143 | 0.027±.001 |
| B_DMPO | 0.0722±.0016 | 0.0894±.002 | 0.083±.001 | 0.016±.000 | 8.316±.110 | 0.015±.000 | 0.0426±.0013 | 0.0625±.0014 | 0.056±.001 | 0.015±.000 | 8.295±.222 | 0.007±.000 | 0.0965±.0010 | 0.1205±.0013 | 0.076±.001 | 0.026±.000 | 8.851±.125 | 0.023±.001 |
| F_PPO | 0.0623±.002 | 0.0792±.0013 | 0.085±.002 | 0.023±.000 | 7.612±.214 | 0.011±.000 | 0.0568±.0009 | 0.0761±.0011 | 0.123±.003 | 0.024±.000 | 7.599±.164 | 0.005±.000 | 0.0968±.0017 | 0.1202±.0017 | 0.083±.002 | 0.028±.001 | 8.795±.103 | 0.022±.000 |
| F_S-DPO | 0.0776±.0009 | 0.0949±.0021 | 0.085±.002 | 0.019±.000 | 8.368±.094 | 0.016±.000 | 0.0640±.0019 | 0.0838±.0017 | 0.075±.002 | 0.016±.000 | 8.435±.086 | 0.007±.000 | 0.1047±.0025 | 0.1262±.0026 | 0.073±.001 | **0.017±.000** | 9.205±.112 | 0.026±.000 |
| F_RosePO | 0.0697±.0013 | 0.0867±.0025 | 0.128±.003 | 0.028±.000 | 8.566±.172 | 0.017±.000 | 0.0603±.0017 | 0.0794±.0022 | 0.307±.005 | 0.059±.001 | 8.459±.189 | 0.008±.000 | 0.1005±.0026 | 0.1207±.0025 | 0.189±.005 | 0.037±.001 | 9.315±.094 | 0.027±.000 |
| F_DMPO | 0.0735±.0008 | 0.0918±.0026 | 0.063±.002 | **0.012±.000** | 8.588±.138 | 0.017±.000 | 0.0648±.0018 | 0.0873±.0025 | 0.043±.001 | 0.013±.000 | 8.274±.094 | 0.007±.000 | 0.0979±.0025 | 0.1217±.0015 | 0.072±.001 | 0.022±.000 | 8.765±.133 | 0.024±.001 |
| C_PPO | 0.0670±.0012 | 0.0834±.0012 | 0.082±.002 | 0.022±.001 | 7.847±.110 | 0.012±.000 | 0.0661±.0020 | 0.0797±.0018 | 0.112±.002 | 0.022±.000 | 8.21±.10 | 0.006±.000 | 0.0978±.0016 | 0.1213±.0026 | 0.082±.001 | 0.027±.000 | 8.763±.100 | 0.023±.001 |
| C_S-DPO | **0.0805±.0012** | **0.0965±.0027** | 0.081±.002 | 0.018±.000 | 8.424±.124 | 0.017±.000 | **0.0737±.0011** | **0.0914±.0012** | 0.067±.002 | 0.014±.000 | 8.765±.171 | **0.009±.000** | **0.1068±.0028** | **0.1265±.0017** | 0.071±.0008 | 0.021±.000 | 9.382±.173 | 0.027±.001 |
| C_RosePO | 0.0717±.0018 | 0.0867±.0020 | 0.125±.004 | 0.027±.000 | **8.832±.159** | **0.019±.000** | 0.0706±.0019 | 0.0883±.0013 | 0.258±.004 | 0.056±.001 | **8.931±.165** | **0.009±.000** | 0.1036±.0016 | 0.1243±.0035 | 0.169±.0032 | 0.037±.001 | **9.537±.200** | **0.029±.000** |
| C_DMPO | 0.0756±.0023 | 0.0928±.0025 | **0.062±.002** | **0.012±.000** | 8.745±.236 | 0.018±.000 | 0.0722±.0014 | 0.0908±.0013 | **0.041±.001** | **0.012±.000** | 8.652±.152 | 0.008±.000 | 0.1002±.0015 | 0.1218±.0031 | 0.071±.001 | 0.022±.000 | 8.936±.260 | 0.028±.001 |

prove, and on-policy training typically converges faster and achieves stronger final performance; overall, combining offline constraints with on-policy updates yields the most effective and stable optimization.

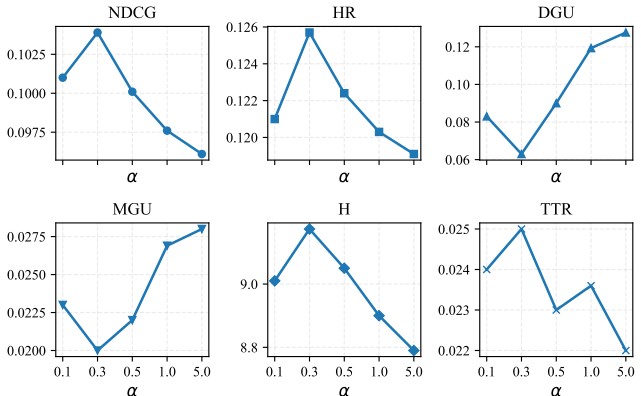

*Figure 4.* Impact of the hyperparameter $\alpha$ on recommendation metrics (NDCG, HR, DGU, MGU, $H$, and TTR). Each subplot shows model performance under different $\alpha$ values, reflecting the trade-offs between recommendation accuracy and diversity.

### 4.6. Robustness and Hyperparameter Sensitivity (RQ5)

We study the robustness of CFlower with respect to the two hyperparameters in the conservative objective (Eq. 9). In particular, $\alpha$ controls the strength of the conservative regularizer, mediating the trade-off between accuracy and exposure balance. Fig. 4 shows that CFlower exhibits clear sensitivity to $\alpha$. Small $\alpha$ yields behavior closer to vanilla SubTB, leaving offline issues (e.g., unsupported-flow leakage) less addressed and limiting exposure gains. As $\alpha$ increases, the conservative term more effectively suppresses unsupported flow mass, yielding better-calibrated exposure. However, excessively large $\alpha$ can lead to noticeable accuracy losses. A moderate $\alpha$ provides the best balance and is used as the default setting.

Apart from $\alpha$, we also evaluated the behavior of $\lambda$ in Eq. 9, where $\lambda > 1$ puts more weight on longer sub-trajectories while $\lambda < 1$ emphasizes shorter ones. In our experiments, $\lambda = 0.9$ achieves slightly better performance. Fig. 5 further indicates mild sensitivity around $\lambda \approx 1$, suggesting that the conservative reweighting is not overly dependent

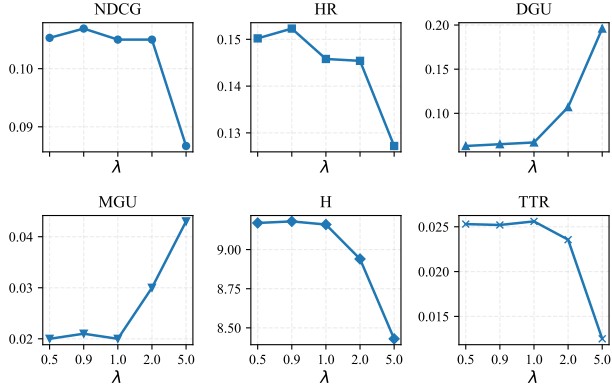

*Figure 5.* Impact of the hyperparameter $\lambda$ on recommendation performance. Each subplot shows model performance under different $\lambda$ values, demonstrating how the focus on long or short trajectories influences various metrics.

on precise length emphasis. Taken together, these results demonstrate that CFlower is robust and can be smoothly tuned to prioritize accuracy or exposure diversity depending on application needs.

## 5. Conclusion

Offline recommender environments pose a fundamental challenge for applying GFlowNet objectives: the learned flow can be underconstrained outside the logged support, leading to distribution mismatch and poor generalization when directly using online-style training. To address this, we proposed CFlower, a conservative GFlowNet framework for LLM-based next-item recommendation that integrates a tree-characterized conservative SubTB objective to suppress unsupported flow mass, together with a dataset-constrained policy learning setting on a dataset-induced DAG that approximates the benefits of on-policy exploration while remaining practical in offline logs. Empirically, across three Amazon domains, CFlower improves distributional matching and delivers a stronger accuracy–exposure trade-off than prior GFlowNet and SFT baselines, and it further serves as a more reliable reference policy for downstream RL fine-tuning.

Despite these gains, our approach inherits limitations of purely offline training: it is bounded by the coverage and biases of logged data, and may still struggle under severe catalog shift or evolving user intent. In future work, we plan to extend CFlower to richer reward designs and counterfactual offline evaluation, and to study scalable training and deployment in more realistic recommendation settings with longer horizons and evolving item catalogs.

## Accessibility

We have made efforts to ensure this paper is accessible to readers with disabilities and sensory or neurological differences. All figures include descriptive captions that explain their content and findings. Tables are structured with clear headers and use text-based formatting (e.g., bold and underline) rather than relying solely on color to convey information. Mathematical notation follows standard conventions and is accompanied by textual explanations. The document structure uses clear section headings and hierarchical organization to facilitate navigation. All implementations are available at https://github.com/yuxuan9982/CFlower.

## Acknowledgements

This paper is partially supported by the National Natural Science Foundation of China (No.12227901), the Natural Science Foundation of China Youth Project (No.62402472), and the Natural Science Foundation of Jiangsu Province of China Youth Project (No.BK20250484). The AI-driven experiments, simulations and model training were performed on the robotic AI-Scientist platform of Chinese Academy of Sciences.

## Impact Statement

This paper presents work whose goal is to advance the field of Machine Learning, specifically in improving fairness and diversity in LLM-based recommendation systems under offline settings. Our method addresses distributional shift and non-identifiability issues in offline LLM-based recommendation via GFlowNet, which can lead to more reliable and equitable recommendation policies.

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

# A. LLM Usage Statement

We only use LLMs as a language optimization tool to polish sentences, improving their readability and fluency. The LLM did not contribute to the scientific ideas, algorithm design, or experimental setup. All substantive content, reasoning, and conclusions are entirely the product of the authors. We accept full responsibility for all content in the paper, including parts refined or corrected by the LLM, and affirm that no text generated by the LLM constitutes original scientific contributions attributed to it.

# B. Related Work

**LLM-based sequential recommendation.** LLM-based recommenders often textualize user histories and predict the next item via supervised fine-tuning (SFT) under logged interactions (Bao et al., 2025; 2024), building on sequential recommendation backbones such as GRU4Rec, Caser, SASRec, and BERT4Rec (Hidasi et al., 2015; Tang & Wang, 2018; Kang & McAuley, 2018; Sun et al., 2019), with growing attention to fairness and bias control (Jiang et al., 2024). Related imbalanced-learning studies on graph-structured data further explore scale-unbiased representation conversion and dynamic balanced prototypes (Wang et al., 2026; Wang et al.), which are aligned with our emphasis on balancing utility/exposure across uneven item groups. Parameter-efficient adaptation techniques for large models, such as token-wise input-output low-rank projections (Li et al., 2025; 2026), are also relevant for practical recommender fine-tuning.

**Process supervision for generative recommendation.** Flower introduces *process supervision* by enforcing flow consistency on token-generation trajectories, providing token-level guidance beyond end-to-end SFT (Gao et al., 2025). This connects to broader lines of work on preference learning and step-level verification/supervision for training generative models (Christiano et al., 2017; Ouyang et al., 2022; Lightman et al., 2023).

**Offline learning for recommender systems.** Because true online evaluation is costly and simulators can be unreliable (Kohavi et al., 2020; Shi et al., 2019; Chandak et al., 2019; Tobin et al., 2017), many methods target offline learning from static logs, which faces partial coverage and distribution shift (Levine et al., 2020; Li et al., 2018). In recommendation, offline evaluation and policy learning are often studied through counterfactual estimation and propensity correction under biased feedback (Swaminathan & Joachims, 2015; Joachims et al., 2017; Schnabel et al., 2016), and conservative regularization is a common strategy to mitigate overestimation in offline RL (Kumar et al., 2020; Fujimoto et al., 2019).

**GFlowNets for recommendation.** GFlowNets (Bengio et al., 2021) have been explored for recommendation in simulated environments (e.g., GFlowGR (Wang et al., 2025)) as well as GFN4rec (Liu et al., 2023) and, more recently, in offline settings where naïvely applying online objectives can cause flow mismatch; COFlownet addresses this via conservative regularization (Zhang et al., 2025). We build on this line with a conservative SubTB objective for token-prefix recommendation.

# C. Proof of Sec. 3.1

Let $\mathcal{G} = (\mathcal{S}, \mathcal{A})$ be a DAG with initial state $s_0$ and terminal set $\mathcal{X} \subset \mathcal{S}$. A SubTB training procedure induces a collection $\mathcal{C}$ of subtrajectory constraints, where each constraint is indexed by a pair $(\tau, i, j)$ with $\tau = (s_0, \ldots, s_T)$ and $0 \le i < j \le T$, and enforces

$$f(s_i) + \sum_{t=i}^{j-1} \log P_F(s_{t+1} \mid s_t) = f(s_j) + \sum_{t=i}^{j-1} \log P_B(s_t \mid s_{t+1}). \tag{12}$$

We say $(P_F, F)$ *satisfies SubTB on $\mathcal{C}$* if Eq. (12) holds for all constraints in $\mathcal{C}$.

Define the set of *constraint-active edges*

$$\mathcal{E}_{\mathcal{C}} = \left\{ (s \to s') \in \mathcal{A} : \exists (\tau, i, j) \in \mathcal{C} \text{ s.t. } (s \to s') \text{ appears in } \tau_{i:j} \right\}. \tag{13}$$

We also define the *constraint connectivity graph* $\mathcal{H}_{\mathcal{C}}$ on states, where $s$ and $s'$ are connected if they co-occur in at least one constraint $(\tau, i, j) \in \mathcal{C}$ (equivalently, both appear in the same subtrajectory segment that is constrained). Let $\{\mathcal{K}_m\}_{m=1}^{M}$ denote the connected components of $\mathcal{H}_{\mathcal{C}}$.

**Proposition** 1 State-Flow Scaling Ambiguity

Assume $(P_F, F)$ satisfies SubTB on $\mathcal{C}$. Let $\alpha : \mathcal{S} \to \mathbb{R}_{>0}$ satisfy $\alpha(s_0) = 1$ and be constant on each connected component $\mathcal{K}_m$ of $\mathcal{H}_{\mathcal{C}}$. Define $\tilde{F}(s) = \alpha(s)F(s)$ and $\tilde{P}_F = P_F$. Then $(\tilde{P}_F, \tilde{F})$ also satisfies SubTB on $\mathcal{C}$.

*Proof.* Fix any constraint $(\tau, i, j) \in \mathcal{C}$ with subtrajectory $\tau_{i:j} = (s_i, \ldots, s_j)$. Because $s_i$ and $s_j$ co-occur in this constraint, they lie in the same connected component of $\mathcal{H}_{\mathcal{C}}$, hence $\alpha(s_i) = \alpha(s_j)$. Write $\tilde{f}(s) = \log \tilde{F}(s) = f(s) + \log \alpha(s)$. Then

$$\tilde{f}(s_i) - \tilde{f}(s_j) = \big(f(s_i) - f(s_j)\big) + \big(\log \alpha(s_i) - \log \alpha(s_j)\big) = f(s_i) - f(s_j).$$

Since $\tilde{P}_F = P_F$ and $P_B$ is unchanged, the forward and backward log-sums along $\tau_{i:j}$ remain identical. Therefore Eq. (12) holds for $(\tilde{P}_F, \tilde{F})$ for this constraint. As the choice of $(\tau, i, j)$ was arbitrary, $(\tilde{P}_F, \tilde{F})$ satisfies SubTB on all constraints in $\mathcal{C}$. $\qquad\square$

**Proposition2** Local Transition Redistribution

Assume $(P_F, F)$ satisfies SubTB on $\mathcal{C}$. Fix a state $s$ and define its constraint-active outgoing set

$$\mathcal{A}_{\mathcal{C}}(s) = \{s' \in \mathcal{S} : (s \to s') \in \mathcal{E}_{\mathcal{C}}\}.$$

Assume $\mathcal{A}_{\mathcal{C}}(s) \neq \emptyset$ and there exists at least one outgoing neighbor $\bar{s}$ of $s$ with $(s \to \bar{s}) \notin \mathcal{E}_{\mathcal{C}}$ (i.e., an outgoing transition of $s$ that never appears in any constraint in $\mathcal{C}$). Then there exists a one-parameter family of modified pairs $(P_F^{(\kappa)}, F^{(\kappa)})$, $\kappa \in (0, 1)$, such that: (i) all SubTB constraints in $\mathcal{C}$ remain satisfied, and (ii) the forward policy at $s$ is altered.

*Proof.* Fix $\kappa \in (0, 1)$. We construct $(P_F^{(\kappa)}, F^{(\kappa)})$ as follows.

*Step 1: modify $P_F$ at state $s$.* For all constraint-active edges $(s \to s') \in \mathcal{E}_{\mathcal{C}}$, define

$$P_F^{(\kappa)}(s' \mid s) = \kappa \, P_F(s' \mid s).$$

Let $\Delta = 1 - \sum_{s' \in \mathcal{A}_{\mathcal{C}}(s)} P_F^{(\kappa)}(s' \mid s)$ be the remaining probability mass at $s$. By construction, $\Delta > 0$ because $\kappa < 1$ and $\mathcal{A}_{\mathcal{C}}(s) \neq \emptyset$. Distribute $\Delta$ arbitrarily over the remaining outgoing neighbors $s''$ of $s$ with $(s \to s'') \notin \mathcal{E}_{\mathcal{C}}$, ensuring that $P_F^{(\kappa)}(\cdot \mid s)$ remains a valid distribution. For all other states $t \neq s$, set $P_F^{(\kappa)}(\cdot \mid t) = P_F(\cdot \mid t)$.

*Step 2: adjust the log-flow at $s$.* Define $f^{(\kappa)}(t) = f(t)$ for all $t \neq s$, and

$$f^{(\kappa)}(s) = f(s) - \log \kappa, \qquad F^{(\kappa)}(s) = \exp(f^{(\kappa)}(s)).$$

*Step 3: verify SubTB constraints.* Consider any constraint $(\tau, i, j) \in \mathcal{C}$ with subtrajectory $\tau_{i:j}$. If $s$ does not appear in $\tau_{i:j}$ as a source state of an edge in $\mathcal{E}_{\mathcal{C}}$, then neither $f$ nor the relevant $P_F$ terms have changed on that subtrajectory, so Eq. (12) remains true.

Otherwise, whenever an edge $(s \to s') \in \mathcal{E}_{\mathcal{C}}$ appears in $\tau_{i:j}$, the forward log-probability term changes by

$$\log P_F^{(\kappa)}(s' \mid s) = \log P_F(s' \mid s) + \log \kappa.$$

At the same time, the subtrajectory includes exactly one occurrence of $f(s)$ as the starting-state term for the segment that begins at $s$ (equivalently, the contribution of $f(s)$ enters with coefficient $+1$ on the LHS and/or with coefficient $-1$ depending on whether $s$ coincides with $s_i$ or $s_j$; crucially, the local SubTB residual involves $f(s)$ and the sum of outgoing logs from $s$ additively). The shift $f^{(\kappa)}(s) = f(s) - \log \kappa$ cancels the added $\log \kappa$ coming from any constraint-active transition out of $s$. Since $P_B$ is unchanged and no other terms are modified, Eq. (12) holds for this constraint. As the constraint was arbitrary, all constraints in $\mathcal{C}$ remain satisfied. $\qquad\square$

**Proposition3** Backward Compensation Invariance (Gauge Symmetry)

Let $(P_F, F)$ satisfy SubTB on $\mathcal{C}$ under a backward policy $P_B$. Let $g : \mathcal{S} \to \mathbb{R}$ be any function with $g(s_0) = 0$. Define transformed quantities:

$$f'(s) = f(s) + g(s), \tag{14}$$
$$\log P'_F(s' \mid s) = \log P_F(s' \mid s) + g(s) - g(s'), \tag{15}$$
$$\log P'_B(s \mid s') = \log P_B(s \mid s') + g(s') - g(s). \tag{16}$$

Then, for every constraint in $\mathcal{C}$, the SubTB equality holds with $(f', P_F', P_B')$ if and only if it holds with $(f, P_F, P_B)$. In particular, SubTB residuals are invariant under this transformation.

*Proof.* Fix any constraint $(\tau, i, j) \in \mathcal{C}$ with $\tau_{i:j} = (s_i, \ldots, s_j)$. Consider the SubTB residual in log form:

$$\mathcal{R}_{i:j} = f(s_i) - f(s_j) + \sum_{t=i}^{j-1} \Big( \log P_F(s_{t+1} \mid s_t) - \log P_B(s_t \mid s_{t+1}) \Big).$$

Under the transformation, we have

$$f'(s_i) - f'(s_j) = f(s_i) - f(s_j) + g(s_i) - g(s_j).$$

For the forward terms,

$$\sum_{t=i}^{j-1} \log P_F'(s_{t+1} \mid s_t) = \sum_{t=i}^{j-1} \log P_F(s_{t+1} \mid s_t) + \sum_{t=i}^{j-1} \big( g(s_t) - g(s_{t+1}) \big).$$

The last sum telescopes to $g(s_i) - g(s_j)$. Similarly, for the backward terms,

$$\sum_{t=i}^{j-1} \log P_B'(s_t \mid s_{t+1}) = \sum_{t=i}^{j-1} \log P_B(s_t \mid s_{t+1}) + \sum_{t=i}^{j-1} \big( g(s_{t+1}) - g(s_t) \big),$$

which telescopes to $g(s_j) - g(s_i)$.

Putting everything together, all $g$-dependent contributions cancel exactly, yielding

$$\mathcal{R}_{i:j}' = \mathcal{R}_{i:j}.$$

Therefore the SubTB residual is invariant for every constrained subtrajectory, and the SubTB equalities hold under $(f', P_F', P_B')$ if and only if they hold under $(f, P_F, P_B)$. $\square$

**Theorem 1.** Non-identifiability of SubTB

Assume there exists at least one non-terminal state and at least two distinct terminal trajectories from $s_0$ to $\mathcal{X}$. Let $\mathcal{C}$ be a non-empty set of SubTB constraints and suppose $(P_F, F)$ satisfies SubTB on $\mathcal{C}$ (under some $P_B$). Then SubTB admits a non-trivial equivalence class of solutions: there exist infinitely many distinct $(P_F^{(\epsilon)}, F^{(\epsilon)})$ that satisfy all constraints in $\mathcal{C}$ while inducing different terminal-state distributions.

*Proof.* We construct a one-parameter family of solutions via the gauge transformation in Proposition 3.

Choose a function $g : \mathcal{S} \to \mathbb{R}$ with $g(s_0) = 0$ that is not constant on all states reachable from $s_0$ (e.g., pick two states $s$ and $t$ with $g(s) \neq g(t)$). For any scalar $\epsilon \in \mathbb{R}$, define $g_\epsilon(s) = \epsilon g(s)$ and apply Proposition 3 to obtain transformed quantities $(f^{(\epsilon)}, P_F^{(\epsilon)}, P_B^{(\epsilon)})$:

$$f^{(\epsilon)}(s) = f(s) + g_\epsilon(s), \tag{17}$$

$$\log P_F^{(\epsilon)}(s' \mid s) = \log P_F(s' \mid s) + g_\epsilon(s) - g_\epsilon(s'), \tag{18}$$

$$\log P_B^{(\epsilon)}(s \mid s') = \log P_B(s \mid s') + g_\epsilon(s') - g_\epsilon(s). \tag{19}$$

By Proposition 3, all SubTB residuals on $\mathcal{C}$ are invariant, hence $(P_F^{(\epsilon)}, F^{(\epsilon)})$ satisfies SubTB on $\mathcal{C}$ for every $\epsilon$.

It remains to show that this family is non-trivial in the sense that the induced terminal-state distribution changes with $\epsilon$. Consider two distinct trajectories $\tau$ and $\tilde{\tau}$ from $s_0$ to (possibly different) terminal states. Their forward-probability ratio under $P_F^{(\epsilon)}$ is

$$\frac{P_F^{(\epsilon)}(\tau)}{P_F^{(\epsilon)}(\tilde{\tau})} = \frac{P_F(\tau)}{P_F(\tilde{\tau})} \cdot \exp \Big( \sum_{(s \to s') \in \tau} (g_\epsilon(s) - g_\epsilon(s')) - \sum_{(s \to s') \in \tilde{\tau}} (g_\epsilon(s) - g_\epsilon(s')) \Big).$$

Each sum telescopes, so the exponent becomes

$$g_\epsilon(s_0) - g_\epsilon(x) - \big(g_\epsilon(s_0) - g_\epsilon(\tilde{x})\big) = g_\epsilon(\tilde{x}) - g_\epsilon(x),$$

where $x$ and $\tilde{x}$ are the terminal states of $\tau$ and $\tilde{\tau}$, respectively. If there exist terminal states $x, \tilde{x}$ with $g(x) \neq g(\tilde{x})$, then the ratio depends on $\epsilon$, implying that the terminal distribution varies with $\epsilon$.

More generally, even when multiple trajectories lead to the same terminal state, the marginal terminal probability $P_F^{(\epsilon)}(x)$ aggregates over trajectories and inherits the $\epsilon$-dependent reweighting induced by $g_\epsilon$ unless the graph is degenerate or $g$ is constant on all reachable terminal states. Therefore, there exist $\epsilon_1 \neq \epsilon_2$ such that $P_F^{(\epsilon_1)}(\cdot) \neq P_F^{(\epsilon_2)}(\cdot)$, establishing a continuum of distinct solutions with identical SubTB constraints. $\qquad\square$

# D. Experiment Details

## D.1. Historical data representation

In the setup of RQ1, we only test GFlowNets' ability to fit a target distribution, thus no user-specific historical data is needed. In the setup of other research questions, we encode each user's historical data into the LLM prompt and train the corresponding logits using SFT and CSubTB loss.

## D.2. Implementation details

For fair comparison, we align our experimental setup with Flower (Gao et al., 2025) whenever applicable. As for CSubTB, Eq. 8 can be equivalently rewritten as $F(s)(1 - \sum_{a \in \mathcal{A}_{\mathcal{D}}(s)} P_F(a \mid s))$ which avoids explicit enumeration over unsupported actions, while the computational overhead of either formulation is negligible compared to other components of training.

