# OpenReview forum: "Improving LLM-Based Recommenders with Conservative Generative Flow Networks"
_ICML.cc/2026/Conference — ICML 2026 regular_

### Official Review · Reviewer_4e59 · 2026-02-26

**Soundness:** 2
**Presentation:** 2
**Significance:** 3
**Originality:** 3
**Overall Recommendation:** 3
**Confidence:** 3

**Summary:**

The paper studies offline learning for GFlowNet-style, LLM-based recommenders and shows that directly applying SubTB on a dataset-induced token-prefix DAG yields non-identifiability and distribution shift due to unsupported transitions. It formalizes three invariances that leave SubTB unsolved under partial support, and proposes a CSubTB objective that penalizes forward flow allocated to unsupported actions, combined with constrained on-policy sampling on the dataset DAG. Across three Amazon domains, the method improves distributional matching and achieves a more favorable accuracy–exposure trade-off than SFT and prior GFlowNet baselines, and also serves as a stronger reference policy for downstream RL fine-tuning.

**Compliance With Llm Reviewing Policy:**

Affirmed.

**Final Justification:**

As for whether the Accessibility section counts as over-length or format non-compliance is best left to the other reviewers and the area chairs to decide. If this is not considered a formatting violation, then my score should be interpreted as Weak Accept.

**Key Questions For Authors:**

1. How are terminal rewards $R(x)$ specified and enforced during CSubTB training for the sequential task? Is there an explicit constraint tying $F(x)$ to $R(x)$, and does this prevent trivial shrinkage of all $F(s)$ to minimize the conservative penalty?
2. Is generation for RQ1 and RQ2 fully constrained to the dataset-induced prefix DAG during evaluation, or are models allowed to produce arbitrary tokens/titles?
3. How precisely are generations mapped to item IDs?

**Limitations:**

Yes

**Strengths And Weaknesses:**

## Strengths

1. The paper targets a emerging field that offline LLM-based recommendation is a practically important regime where safety/fairness constraints and lack of exploration are the norm, offering a principled GFlowNet treatment and stabilizing objective is impactful.
2. It states clearly that three distinct sources of non-identifiability for SubTB under partial support in a unified way, helping clarify why offline transfer of online GFN objectives can fail.
3. The constrained on-policy sampling within the dataset-induced DAG is a practical and sensible design that respects offline constraints while exploiting model guidance.
4. The method is further validated as a reference policy for several downstream RL-style fine-tuning procedures, which strengthens the claim of improved generalizability and stability.
5. The empirical study covers three realistic Amazon domains, reporting both recommendation accuracy and several exposure or diversity metrics, plus a distributional-matching analysis on unconditional item frequencies.

## Weaknesses

1. The main text includes a **Section Accessibility** that appears to push the manuscript beyond the page limit. I checked the ICML author guidelines but could not find a clear requirement for including it. I don't know whether this should trigger an immediate desk rejection due to length non-compliance.
2. The non-identifiability proof of Thm. 1 leverages a gauge transformation that alters both $P_F$ and $P_B$, yet the main recommender instantiation assumes a fixed or trivial backward policy on a tree, i.e., unique parent. It is unclear how much of the identified invariance persists when $P_B$ is fixed and the structure is a strict prefix tree beyond state-flow scaling and local redistribution.
3. The specification of reward $R(x)$ in Sec. 3 is under-explained for the sequential setting, leaving unclear how terminal flows are anchored to data or reward during CSubTB training.
4. The relationship between the proposed constrained on-policy sampling and the on-policy process-supervision used in Flower is not highlighted, so readers may perceive overlap without a clear positioning.

---

> ### Author Rebuttal · Authors · 2026-03-31
>
> Dear Reviewer 4e59,
> Thank you very much for your valuable comments, which are crucial to the improvement of our paper. We would like to clarify your concerns point by point in the following.
>
> > W1.Format issue.
>
> Thank you very much for pointing this out. We apologize for any confusion caused by the Accessibility section. Our intention was simply to provide an accessibility-oriented statement for readers, rather than to claim extra space for technical content. The Accessibility part is included in the ICML template itself and appears there as an unnumbered \section*{Accessibility} section. We certainly do not wish to risk any non-compliance with the ICML formatting rules. If the section is not appropriate to keep in the main paper, we would be very happy to move it to the appendix or supplementary material and follow whatever format the chairs consider most appropriate. We hope this can be treated as a straightforward formatting adjustment rather than a substantive attempt to exceed the page limit, and we are fully willing to make the corresponding change immediately if requested.
>
> > W2. Non-identifiability proof of Thm. 1. Identified invariance when $P_B$ is fixed.
>
> Thank you for raising this point. In thm1, we want to prove that all learnable parameters of GFlowNet are non-identifiable. When $P_B$ is learnable, the triple $(P_F, F, P_B)$ exhibits non-identifiability. However, when the state space forms a tree (where each state has a unique parent), $P_B$ becomes deterministic and need not be learned—it is fully determined by the tree structure. In this case, only $(P_F, F)$ remain non-identifiable and the invariance related to $P_B$ is no longer presented. However, the set of non-identifiable parameters is still **all** learnable parameters: the gauge transformation's effect are precisely the degrees of freedom that remain unidentifiable regardless of whether $P_B$ is fixed or learned. Formally, the non-identifiability caused by state-flow scaling and local transition redistribution remains. There still exists a continuous family of valid $(P_F, F)$ pairs, meaning the gauge orbit sweeps through all learnable parameters non-trivially. Hence no learnable parameter escapes the transformation, and identifiability fails for the full learnable parameter set in both cases.
>
>
> > W3. The specification of reward R(x).
>
> Thank you for pointing this out. We agree that Sec. 3 did not explain the reward instantiation clearly enough in the sequential setting. In our setup, a complete sequence corresponds to a movie title, and the base terminal reward is derived from the empirical frequency of that title in the data. To incorporate personalization, we further follow Flower variant (1), where user-specific preference information enters through the process reward via the auxiliary score $p_{ui}$. Thus, terminal flows are anchored by the data-derived reward over complete titles, while personalization is introduced separately through process-level reward shaping. We will revise Sec. 3 to make this distinction explicit.
>
> > W4. The relationship between the proposed constrained on-policy sampling and the on-policy process-supervision used in Flower is not highlighted, so readers may perceive overlap without a clear positioning.
>
> Thank you for this helpful comment. We agree that the relationship to Flower should be positioned more explicitly. Flower is fundamentally an online method, so its use of on-policy process supervision is natural in that setting. By contrast, our focus is on an offline recommendation setting based on logged data, where prior work is more commonly framed from an off-policy perspective. From this viewpoint, our goal is not simply to reuse Flower’s online formulation, but to study how related process-supervision ideas can be adapted to an offline, constrained, recommender-specific setting. We will revise the paper to make this distinction clearer in the related work and method overview.
>
> > Q1. Definition of R(x). Whether tying F(x) to R(x), whether prevent shrinkage of all F(s).
>
> Thank you for this insightful comment. As discussed in W3, there is indeed an explicit constraint tying F(x) to R(x) for all terminal states. And your observation that this prevent trivial shrinkage of all F(s) is right, for more details of the shrinkage of F(s), please refer to out response to`W1` of reviewer Gv3A.
>
> > Q2. Generation can produce arbitrary tokens/titles or not？
>
> Thank you for this helpful comment. The generation is constrained by prefix-based token masking during decoding—implemented via `prefix_allowed_tokens_fn`. For more details, please refer to out response to`W1` of reviewer 17Cf.
>
> > Q3. How precisely are generations mapped to item IDs?
>
> Thank you for this question. The prefix tree is constructed from the item-title vocabulary, so a valid completed generation corresponds to a terminal title in this vocabulary. Each such terminal title is associated with its corresponding item ID via a fixed lookup table.

---

> > ### Author Rebuttal · Reviewer_4e59 · 2026-03-31
> >
> > I have no further technical questions.
> >
> > As for whether the Accessibility section counts as over-length or format non-compliance is best left to the other reviewers and the area chairs to decide. If this is not considered a formatting violation, then my score should be interpreted as **Weak Accept**.

---

> > > ### Author Response · Authors · 2026-04-03
> > >
> > > Thank you very much for considering our paper as Weak Accept, which is a tremendous encouragement to our work. Thank you for your recognition of the significance and originality of our work. We are glad that we fully resolved your concerns. We would be grateful if you could kindly consider reflecting your updated assessment in the score (moving to 4-Weak Accept).
> > >
> > > Regarding the formatting issue, we would like to provide a further clarification. The `Accessibility Section` is part of the official ICML 2026 template (downloaded directly from https://media.icml.cc/Conferences/ICML2026/Styles/icml2026.zip). In the template, `Accessibility Section` appears alongside the `Impact Statement` using the same `\section*` command, indicating that both are outside the main content of the paper. Following the instruction in the template *"Authors are kindly asked to make their submissions as accessible as possible for everyone including people with disabilities and sensory or neurological differences"*, we put `Accessibility Section` outside the main text, and alongside the `Impact Statement` as in the template. The purpose of this was merely to provide a link to our code. We absolutely have no intention of violating the ICML policy regarding the length of the main text. Actually, this section could instead be condensed into a single anonymous URL placed within the abstract, which would have no effect whatsoever on the layout or length of the main manuscript. This means our submission occupies no more space in the main content than any other submission, and therefore raises no fairness concerns with respect to other papers. We will also explain the issue with Area Chair and possitively make any adjustment in the revised version if needed.

---

### Official Review · Reviewer_Gv3A · 2026-03-05

**Soundness:** 3
**Presentation:** 3
**Significance:** 3
**Originality:** 3
**Overall Recommendation:** 5
**Confidence:** 4

**Summary:**

This paper focuses on addressing a critical gap in existing research: why GFlowNets often fail to learn the correct distribution when applied to LLM-based recommenders. Specifically, the authors analyze why the standard Sub-Trajectory Balance (SubTB) objective becomes non-identifiable when applied to dataset-induced token-prefix DAGs. Through this analysis, they identify three key invariance mechanisms that undermine SubTB’s performance: flow overestimation caused by disconnected-component scaling, local forward mass redistribution, and a gauge-like compensation effect involving forward flows, backward flows, and overall flow. To mitigate these issues, the authors propose a conservative variant of the SubTB objective (CSubTB), which introduces a penalty for unsupported forward flows. They further pair this objective with constrained on-policy sampling conducted directly on the dataset DAG. Experiments conducted on three Amazon domains validate the effectiveness of this approach: compared to SFT and other existing GFlowNet baselines, the proposed method achieves better distributional matching and strikes a more favorable balance between recommendation accuracy and exposure. Additionally, the results demonstrate that CFlower serves as a more robust reference policy for downstream reinforcement learning fine-tuning.

**Compliance With Llm Reviewing Policy:**

Affirmed.

**Final Justification:**

my concerns have been addressed

**Key Questions For Authors:**

- I have observed that GFlowNets employ a variety of objectives, such as FM and QM in COFlowNet. Why is SubTB chosen as the objective for GFlowNets? Is it feasible to apply other objectives like FM and QM to LLM-based recommenders? Are these objectives effective in such settings?
- For other questions, please see Weaknesses.

**Limitations:**

yes

**Strengths And Weaknesses:**

**Strengths :**

- The paper clearly distinguishes between offline and online recommendation settings, providing a solid conceptual foundation. It formally formalizes offline learning for GFlowNets within LLM-based recommenders and presents a rigorous analysis of the SubTB objective.
- The theoretical proofs are insightful, clearly structured. Figure 1 is particularly clear and illustrative, effectively explaining why vanilla GFlowNets struggle to learn the correct distribution under LLM-based policies.
- Extensive evaluations on three real-world datasets demonstrate that the proposed method improves both ranking accuracy and exposure/diversity metrics. Experiments with five different random seeds further verify the robustness of the approach.
- The work addresses a critical gap at the intersection of LLM-based recommendation and GFlowNets, a highly active and impactful research direction for the ICML community, making it both timely and theoretically significant.

**Weaknesses:**

- The penalty term scales with F(s), yet flows themselves suffer from scaling ambiguities (as noted in Proposition 1). Additional discussion is needed on how the method avoids trivial solutions such as downscaling flows, as well as on any auxiliary normalization or reward anchoring used to stabilize absolute flow magnitudes.
- Evaluation protocols are under-specified. It remains unclear whether next-item metrics rely on full-catalog ranking or sampled negatives, how candidate sets are constructed, and whether train/test splits prevent temporal leakage. These design choices can substantially alter experimental conclusions.
- The RL fine-tuning pipelines (e.g., reward or preference sources for S-DPO/RosePO/DMPO under offline settings) would benefit from more targeted citations and additional methodological details.

---

> ### Author Rebuttal · Authors · 2026-03-31
>
> Dear Reviewer Gv3A,
> Thank you very much for your valuable comments, which are crucial to the improvement of our paper. We would like to clarify your concerns point by point in the following.
> > W1. How the method avoids trivial solutions such as downscaling flows, reward anchoring used to stabilize absolute flow magnitudes.
>
> Thank you for raising this important point. We agree that, in isolation, the conservative penalty scales with \(F(s)\) and therefore should not be interpreted as a standalone mechanism for fixing the absolute scale of flows. In our formulation, the relevant scale is anchored by the terminal reward constraints, since **terminal flows** satisfy $F(x)=R(x)$, while the conservative term is introduced to penalize unsupported forward flow mass. Therefore, the method is not relying on the penalty alone to prevent trivial downscaling; rather, it is the combination of reward-anchored terminal constraints, SubTB/CSubTB flow-matching constraints, and the support-aware penalty that restricts such degenerate directions. In other words, CSubTB is support-aware: it discourages leakage to unsupported actions, while **reward anchoring at terminal states** provides the reference scale for the learned flows. We will clarify in the revision that the conservative term reduces unsupported leakage, rather than serving as an independent global scale-fixing mechanism.
>
> > W2. Evaluation protocols are under-specified. full-catalog ranking or sampled negatives. how candidate sets are constructed. Whether prevent temporal leakage.
>
> Thank you for pointing this out. We reviewed the current manuscript carefully. The paper does state that the datasets are chronologically split with an 8\:1\:1 ratio after time-based truncation and standard preprocessing, which means the train/validation/test partition respects temporal order and is intended to avoid using future interactions during training. We agree that the manuscript does not currently make sufficiently clear whether next-item evaluation is performed over the full item catalog or over a sampled candidate set. We will include additional content in our revision:"our evaluation does not use sampled negatives. Instead, it uses constrained decoding over the valid item-token set, implemented through prefix_allowed_tokens_fn, so the model predicts within the admissible item vocabulary rather than ranking against a separately sampled candidate subset."
> > W3.The RL fine-tuning pipelines would benefit from more targeted citations and additional methodological details.
>
> Thank you for this helpful suggestion. We agree that the current manuscript does not provide enough methodological detail for the downstream offline fine-tuning pipelines. After revisiting our implementation, we can clarify several points more explicitly. First, all downstream fine-tuning methods in RQ3 operate on offline data loaded from fixed train/eval JSON files. For the DPO-style methods, the optimization is implemented with a policy model and a fixed reference model, and the loss is computed from the difference between policy and reference log-probabilities on preferred vs. dispreferred responses. In particular, our S-DPO and DMPO implementations use one chosen response together with multiple rejected responses per prompt. Second, for PPO-style offline fine-tuning, the reward is not obtained from online interaction. Instead, the scalar reward is constructed offline from precomputed recommendation signals, specifically using item-level scores derived from precomputed SASRec logits together with normalization over the candidate item space.
> > Q1.Is it feasible to apply other gfn objectives to LLM-based recommenders?
>
> Thank you for this insightful observation. While GFlowNets can indeed be trained with different objectives (e.g., FM, TB), not all of them are equally suitable for LLM-based recommenders. In our setting, the model is naturally parameterized as an autoregressive forward policy over token/item sequences, rather than as an explicit estimator of edge flows \(F(s,a)\). Objectives such as FM and QM typically require direct parameterization or explicit access to state/action flow quantities, which is substantially less natural and more difficult to scale in an LLM-based recommendation framework.
> In contrast, DB, TB, and SubTB are all compatible with policy-style parameterization. We originally chose SubTB because it offers a favorable trade-off between optimization stability and trajectory-level credit assignment in our setting. Following the reviewer’s suggestion, we additionally evaluated DB and TB under the same experimental setup. The table is given at the [anonymous link](https://anonymous.4open.science/r/16739-anonymous-table-figs-AF52/Table-TB-SubTB.csv). The results show that DB/TB are indeed feasible in LLM-based recommendation and achieve performance comparable to SubTB, while SubTB remains slightly better overall. We will include this additional comparison and discussion in the revised paper.

---

> > ### Author Rebuttal · Reviewer_Gv3A · 2026-04-01
> >
> > The authors have provided counter-arguments and excellent practical solutions to the limitations.

---

> > > ### Author Response · Authors · 2026-04-03
> > >
> > > Thank you very much for your careful review and positive feedback. We greatly appreciate that you are satisfied with our counter-arguments and the practical solutions we provided to address the limitations of our work. We are glad that all your concerns have been fully resolved.

---

### Official Review · Reviewer_knev · 2026-03-09

**Soundness:** 2
**Presentation:** 2
**Significance:** 2
**Originality:** 2
**Overall Recommendation:** 3
**Confidence:** 3

**Summary:**

This paper studies offline GFlowNet training for LLM-based recommendation under partial support from logged data. It argues that standard SubTB training can leak probability mass to unsupported transitions, and proposes CFlower, which adds a conservative penalty on unsupported forward flow together with dataset-constrained on-policy sampling on the logged prefix DAG. The experiments show improvements over Flower on distribution matching and several recommendation metrics.

**Compliance With Llm Reviewing Policy:**

Affirmed.

**Final Justification:**

I have read the authors’ rebuttal, but I find their claims regarding the distinction from offline reinforcement learning somewhat too categorical. While the two settings are not identical, the forward mass leakage issue in GFlowNets appears closely related in spirit to the Q-value overestimation problem in offline reinforcement learning: both can be viewed as arising from estimation errors on poorly supported regions of the space.

Moreover, I am not fully convinced that forward mass leakage or flow overestimation is a practically severe issue. When the sample size is sufficiently large, or when appropriate pruning is applied at inference time, its effect may be substantially mitigated and could become negligible in practice. As an analogy, LLMs also assign nonzero probability to invalid utterances, yet this probability is often so small that it does not materially affect their practical usefulness.

Therefore, I maintain my original assessment.

**Key Questions For Authors:**

1. How does the advantage of CFlower over Flower change as support coverage increases?
   A strong answer would clarify whether the method addresses a broad issue or mainly sparse-support settings.

2. Why are policy-optimization style baselines not included, given that the paper explicitly emphasizes constrained on-policy learning?
   A strong answer would help determine whether the gains are specific to the proposed conservative GFlowNet design.

3. Are all LLM-based baselines built on the same base model and comparable model capacity?
   A strong answer would improve confidence in the fairness of the empirical comparison.

4. Can the authors provide a full per-experiment hyperparameter specification and explain how alpha and lambda were selected?
   A strong answer would improve reproducibility and help assess robustness.

5. How do the authors interpret the cases where SASRec achieves better exposure metrics than CFlower?
   A strong answer would make the empirical trade-offs much clearer.

**Limitations:**

yes

**Strengths And Weaknesses:**

Strengths

1. The paper identifies a meaningful failure mode of offline GFlowNet training in LLM recommendation and formulates it clearly in terms of partial support.

2. The method is conceptually clean, because the conservative penalty and constrained on-policy sampling both directly target unsupported transitions.

3. The theoretical discussion of non-identifiability under partial support makes the paper more principled than a purely empirical regularization tweak.

4. The experiments show that the proposed method consistently improves over Flower on several key metrics.

Weaknesses

1. The method seems somewhat incremental, since its main novelty over Flower is essentially to penalize unsupported flow and restrict optimization to the feasible set defined by the offline data.

2. The paper does not establish the regime in which the method is actually needed, because all of its motivation depends on partial support while it never studies whether the gains disappear when support coverage becomes sufficiently good.

3. The baseline comparison is incomplete and somewhat selective, because the paper stresses constrained on-policy learning without comparing to stronger policy-optimization style alternatives, and it does not explicitly discuss that SASRec is better than CFlower on some exposure-oriented metrics.

4. The experimental setup is not reported clearly enough, since the paper does not clearly specify the base LLM used for each LLM baseline, does not provide a comprehensive table of training hyperparameters across experiments, and gives only limited discussion of how alpha and lambda are chosen in practice.

---

> ### Author Rebuttal · Authors · 2026-03-31
>
> Dear Reviewer knev,
> Thank you very much for your valuable comments, which are crucial to the improvement of our paper. We would like to clarify your concerns point by point in the following.
>
> > W1: Novelty concern.
>
> We thank the reviewer for this comment and would like to clarify our contribution. This work is not merely to add a penalty on unsupported flow or to heuristically restrict optimization to the feasible set. **Our main contribution is to theoratically identify and formalize an offline-specific failure mode that is not addressed in Flower**: under limited offline support, the original objective can assign flow incorrectly to unsupported transitions, leading to issues such as flow overestimation, forward mass leakage, and backward compensation. The analysis can be also generalized to other scenarios.
> The theoratical contribution should be emphasized. Therefore, the proposed conservative correction is not an arbitrary regularizer, but a principled modification of the learning objective to make flow learning better aligned with what is identifiable from offline data. In this sense, the novelty of CFlower lies less in introducing a new framework from scratch, and more in showing that the original objective becomes mis-specified in the offline regime and in deriving a targeted correction for this setting.
>
>
> > W2 and Q1: The regime; Whether gains disappear with coverage.
>
> We thank the reviewer for raising this important point. Our method is specifically designed for the whole offline regime, where unsupported transitions are unavoidable and can distort flow learning. We agree that, in principle, if support coverage became sufficiently rich, the benefit of the conservative correction could diminish. However, this is impossible to arise in realistic LLM-based recommendation settings. The action space is combinatorially large: for example, the vocabulary size of Qwen-2.5 exceeds 150,000 tokens. Under such a space, it is practically infeasible for offline data to provide dense coverage. Therefore, partial support is not a corner case but the practically relevant regime, and CFlower is designed precisely for this setting.
> Furthermore, to reduce your concern, we statistically analyzed the average support coverage of the given three dataset: Movies_and_TV $0.0008\%$ Video_Games $0.00075\%$ CDs_and_Vinyl $0.00081\%$. We also add an additional experiment with different offline data size(10%,50%,100%) as COFlowNet. The results are given in data_size.csv and support-coverage.csv of our [anonymous link](https://anonymous.4open.science/r/16739-anonymous-table-figs-AF52/).
>
> > W3 and Q2,Q5: Comparison to policy-optimization; Why SASRec better on some metrics.
>
> We appreciate the constructive feedback on baseline completeness. We clarify:
> - Policy optimization comparison:
> A critical clarification: policy optimization methods (PPO, DPO, etc.) are typically applied *after* SFT, not during it. Comparing CFlower (SFT-stage method) directly with RL methods would be methodologically inconsistent. Instead, in RQ3, we properly evaluate CFlower's value as a reference policy for downstream RL. Results show CFlower-referenced RL consistently outperforms alternatives: S-DPO with CFlower achieves NDCG 0.0737 on Video Games (vs. 0.0640 with Flower). This demonstrates CFlower's utility across the full pipeline.
> - Sasrec comparison:
> While SASRec shows higher H and TTR, this reflects fundamental differences between traditional ID-based and LLM-based recommendation. Traditional methods like SASRec operate over a discrete item ID space, directly selecting from a fixed candidate set. This enables **arbitrary exposure allocation**: the model can freely assign any probability mass to any item ID, independent of other items' properties. In contrast, LLM-based recommenders generate item representations as token sequences through autoregressive generation. The token vocabulary is shared across all items, creating inherent coupling: tokens in popular items' titles have higher base frequencies in training data. When generating, the LLM's representations naturally cluster semantically similar items, making it harder to achieve uniform exposure across diverse items without sacrificing generation quality. Traditional ID-based methods have no such constraint: they can assign equal probability to any two items regardless of semantic similarity. We regard this gap as a fundamental distinction between traditional and LLM-based recommenders.
>
>
> > W4 and Q3,Q4: The base LLM used for each LLM baseline; A comprehensive table of training hyperparameters across experiments.
>
> We appreciate the feedback on experimental clarity. We acknowledge the omission of base LLM specification. All LLM-based baselines use the same base LLM: Qwen2.5-1.5B-Instruct. Besides, we provide a comprehensive table of hyperparameters in our [anonymous link](https://anonymous.4open.science/r/16739-anonymous-table-figs-AF52/). We will explicitly state this in the revised paper.

---

> > ### Author Rebuttal · Reviewer_knev · 2026-04-01
> >
> > Thank you for the clarification. I appreciate your effort in explaining the motivation and positioning of the method. My concern, however, is not only about the novelty at the methodological level. More importantly, the underlying motivation—such as partial support, distribution shift, and the need for conservative learning under offline settings—has also been explored in related LLM and offline RL literature. In this context, I still find it unclear whether the paper has sufficiently demonstrated the specific advantage of the proposed formulation over other methods with similar motivation.

---

> > > ### Author Response · Authors · 2026-04-03
> > >
> > > We thank the reviewer for this follow-up. Here, we respectfully maintain that our work addresses a **fundamentally different** problem from **existing offline RL-based LLM recommenders**, for the following reasons.
> > > > Differences between flow-based and non-flow-based RL methods
> > >
> > > We would like to first clarify the fundamental distinction between GFlowNet-based (flow-based) methods and conventional non-flow-based approaches in LLM recommenders. Traditional RL methods are typically applied during **a post-SFT alignment(RLHF)** phase, with the objective of maximizing the reward of generated items — for instance, maximizing the score of chosen responses over rejected ones. This reward-maximization objective is inherently aggressive and prone to diversity collapse, which is why such methods require a reference model(KL or inverse KL) from the SFT phase to constrain the policy. GFlowNets, by contrast, are designed with an entirely different objective: rather than maximizing reward, they learn to sample candidates with probability proportional to reward, which is precisely what enables diversity. GFlowNets are applied **during the fine-tuning phase** to directly address the popularity bias and diversity limitations inherent in standard SFT-based recommenders. The goals, mechanisms, and failure modes of these two paradigms are **fundamentally different**, and this distinction is central to understanding the novelty of our work.
> > >
> > > > Conceptual differences of similar terminology
> > >
> > > As noted in `lines 102–109` of our manuscript, the ***distributional shift*** problem we study is **conceptually distinct** from that encountered in conventional offline RL algorithms such as CQL and BEAR. The key distinction lies in the learning objective: traditional RL methods aim to **maximize** the reward of generated candidates, whereas GFlowNets are trained to sample candidates with probability **proportional** to their reward by learning flow consistency. This difference in objective gives rise to entirely different forms of distributional shift. In traditional offline RL, distributional shift manifests as **over-estimation of action values** for out-of-distribution states — a well-studied phenomenon that CQL and BEAR are specifically designed to mitigate. In contrast, the distributional shift in GFlowNets stems from the misalignment between flows and mass leakage, as analyzed in detail in our manuscript. Despite sharing similar terminology, these two phenomena are fundamentally different in nature and should **not be conflated** — which is precisely why we explicitly clarify in our manuscript: "We use distributional shift in a GFlowNet-specific sense…". Furthermore, we emphasize that *partial support* is an observation, not a design motivation. We empirically observed that even after extensive training, the learned distribution remains misaligned with the target distribution. Through experimental and theoretical analysis, we identify unsupported transitions as **a primary source of this misalignment**, which naturally motivates suppressing them to **recover the correct distribution**. Since the distributional shift we study is GFlowNet-specific, the notion of partial support is likewise a **flow-specific concept**, distinct from the motivation behind conservative learning in conventional offline RL.
> > >
> > > > Reviewer's quetion: The specific advantage of the proposed formulation over other methods with similar motivation
> > >
> > > Actually, offline learning is far more critical for **flow-based** methods than for non-flow-based RL approaches in LLM-based recommenders. In flow-based settings, the learned distribution directly determines the quality of the resulting reference model. Any distributional shift between the training data and the learned distribution can lead to severe **misestimation** of the trained policy, which in turn corrupts the entire generative process. This is fundamentally different from non-flow-based offline RL methods (e.g., DPO, SDPO), where **over-estimation** of action values does not catastrophically destabilize the policy—largely and there exists a **reference model** constraints that keep the policy well-behaved.
> > > Given this critical importance of offline learning in the flow-based paradigm, we specifically address the challenge of distributional shift in flow-based offline recommenders. To the best of our knowledge, **no prior work** has investigated this problem, applying GFlowNets to LLM-based recommenders for diversity enhancement under offline settings is itself a **novel** problem formulation. Therefore, both our motivation and theoretical analysis and methodology represent original contributions to the field.
> > >
> > > For above reasons, we respectfully maintain that what we study: partial support, distribution shift, and conservative flow-based learning, are novel and of great importance to offline Recommenders. And our contributions are multi-dimentional: problem formulation, theoretical grounding, and method adjustments.

---

### Official Review · Reviewer_17Cf · 2026-03-15

**Soundness:** 2
**Presentation:** 2
**Significance:** 2
**Originality:** 2
**Overall Recommendation:** 4
**Confidence:** 4

**Summary:**

This paper proposes CFlower, which uses a conservative sub-trajectory balance objective to penalize unsupported token transitions.\
Overall, the authors discuss a major issue regarding distributional shift and provide a technically sound framework supported by extensive experiments across multiple metrics.\
However, the necessity of the GFlowNet approach remains under-justified, as the key question investigated by the art may already be addressed by existing indexing schemes or constrained decoding.

**Compliance With Llm Reviewing Policy:**

Affirmed.

**Final Justification:**

Since the rebuttal addressed most of my concerns, I have raised my score from weak reject to weak accept.

**Key Questions For Authors:**

Q1: How does the proposed CSubTB objective perform compared to simple output vocabulary constraints or prefix trees (trie-based decoding) that naturally prevent the "unsupported transitions" shown in Figure 1?

Q2: Can you provide a more intuitive explanation or a dedicated ablation study showing why GFlowNet's reward-proportional sampling is more effective for diversity than standard LLM sampling techniques?

**Limitations:**

yes

**Strengths And Weaknesses:**

**Strengths**

S1: Comprehensive Evaluation
- The authors evaluate the model across multiple real-world Amazon datasets (CDs, Video Games, Movies) using a wide array of metrics for both recommendation accuracy (NDCG, HR) and distributional quality (Jaccard, KL divergence, TV distance).

S2: Theoretical Grounding
- The paper provides a formal analysis of the failure modes of SubTB in offline settings, identifying three specific sources of non-identifiability: flow overestimation, forward mass leakage, and backward compensation.

S3: Extensive Baselines
- The experimental section includes a robust set of comparisons against traditional sequential models (SASRec) and modern LLM-based approaches, including SFT (BIGRec) and existing GFlowNet methods (Flower)

**Weaknesses**

W1: Motivation for "unsupported transitions"
- The paper seems to overlook standard engineering solutions.\
Specifically, the "invalid recommendations" can often be prevented using an indexing approach [A] or constrained decoding with trie-based indexing schemes (e.g., "prefix_allowed_tokens_fn" argument in transformers library).\
[A] Uncertainty Quantification and Decomposition for LLM-based Recommendation, WWW 2025

W2: Unclear advantage over SFT
- While GFlowNets are intended to improve diversity by sampling proportional to rewards, the paper does not sufficiently explain why this specific framework is superior to SFT-based recommenders that use temperature scaling or diverse beam search.
- Why it is inherently better at generating "diverse" recommendations in a practical sense?

W3: Marginal improvement in accuracy
- Although the model shows better distributional matching, the improvements in traditional accuracy metrics like NDCG appear relatively modest compared to SFT baselines.

---

> ### Author Rebuttal · Authors · 2026-03-31
>
> Dear Reviewer 17Cf,
> Thank you very much for your valuable comments, which are crucial to the improvement of our paper. We would like to clarify your concerns point by point in the following.
>
> > W1: Motivation for "unsupported transitions". Overlook standard engineering solutions like indexing approach [A] or constrained decoding with trie-based indexing schemes.
>
> We would like to first clarify that the motivation for "unsupported transitions" is not to ensure valid generations, but to learn the right distribution. We observed that even after extensive training, the learned distribution is still shifted. Theoretical analysis shows that unsupported transitions are a major source of such shift. Our goal is to address them so as to learn the right distribution within the valid support, rather than merely mask invalid tokens.
>
> Meanwhile, engineering solutions is actually included in our implementation (except for the simple RQ1 demonstration setting), and is also used in the Flower baseline. Specifically, both methods apply prefix-based token masking during decoding, including implementation via `prefix_allowed_tokens_fn` and a constrained logits processor (`CFEnhancedLogitsProcessor` in our code at `train_sft-gfn_logp_div_s_cfn.py`), so that only tokens corresponding to valid item prefixes can be generated. Such implementation is commonly used in LLM-based recommendation systems. Therefore, we did not emphasize the components in our manuscript.
>
> > W2: Unclear advantage over SFT. Superiority to temperature scaling or diverse beam search. Why better at diversity.
>
> Thank you for the thoughtful question. We would like to clarify that CFlower is not a replacement for temperature scaling or diverse beam search, but complementary to them.
>
> Temperature scaling and diverse beam search are decoding-time heuristics, which are operating on a learned distribution. In contrast, CFlower modifies the training objective, directly learning a distribution that balances accuracy and diversity. Combining CFlower with temperature and beam search is simple and technically reasonable.
>
> We would likt to further explain the differences between SFT and CFlower, as both are training methods. SFT trains models through comparing generated output with groundtruth, which aims at squeezing probability mass into the trajectory defined by groundtruth and inherently inherits popularity bias from the dataset. Decoding tricks (temperature scaling and diverse beam search) cannot fully correct this since they do not alter the underlying distribution. GFlowNet-based methods instead shape the entire probability distribution, allocating mass proportional to **reward**, which enables global diversity and exploration control. CFlower explicitly reweights the distribution during training, achieving a better accuracy–diversity trade-off.
>
>
> > W3: Marginal improvement in accuracy.
>
> We are sorry for the reviewer's confusion about the improvement. Actually, the percentage of performance improvement is relatively high. First, compared to its base model BigRec(the SFT version of CFlower), CFlower improves NDCG from 0.0328 to 0.0646 on Video Games (+97%) and from 0.0574 to 0.0737 on CDs & Vinyl (+28%), which are substantial gains. These improvements are achieved while simultaneously improving diversity-related metrics (e.g., DGU, MGU, H, TTR), indicating a strictly better accuracy–diversity trade-off rather than a marginal accuracy gain. Overall, the improvements are both large in magnitude and consistent across datasets, and thus cannot be considered marginal.
>
> > Q1: How does the proposed CSubTB objective perform compared to simple output vocabulary constraints or prefix trees?
>
> As discussed in `W1`, prefix-based token masking is already implemented in both CFlower and Flower to ensure that only valid items are generated. Noting that Flower still suffers from distributional shift even prefix-based token masking is used, the gains of our framework come from improved distribution learning rather than output constraints.
>
> > Q2: Ituition of why CFLower is more effective for diversity?
>
> Intuitively, the key difference is that standard LLM sampling methods (e.g., temperature scaling or beam search) only perturb a fixed model distribution, while GFlowNet explicitly learns a target distribution proportional to rewards.
>
> In SFT-based models, the learned distribution is inherently skewed toward high-frequency (popular) items due to likelihood maximization. Increasing temperature can flatten this distribution, but it does so indiscriminately, introducing randomness without correcting the underlying bias. In contrast, GFlowNet (and Flower/CFlower) enforces that the generation probability satisfies $p(x) \propto\ R(x)$ which directly aligns the model distribution with a desired reward distribution. Therefore, GFlowNet’s advantage is not merely “more randomness,” but learning the correct distribution, which leads to more meaningful and stable diversity in practice.

---

> > ### Author Rebuttal · Reviewer_17Cf · 2026-04-02
> >
> > Since the rebuttal addressed most of my concerns, I have raised my score from weak reject to weak accept.

---

> > > ### Author Response · Authors · 2026-04-03
> > >
> > > Thank you very much for your careful review and constructive comments. We highly appreciate that you have carefully considered our rebuttal and found our responses adequate to address your concerns. We are grateful for your **positive feedback** and the **score adjustment** from weak reject to weak accept.

---

### Decision · Program_Chairs · 2026-04-30

**Decision:**

Accept (regular)

**Comment:**

This paper studies offline GFlowNet training for LLM-based recommendation, identifies the partial-support failure of SubTB under dataset-induced token-prefix DAGs, and proposes CFlower to improve distributional matching and the accuracy–exposure trade-off.

There are still some open questions about how much of the gain is unique to the proposed conservative GFlowNet formulation, and about the clearest empirical characterization of the regime where the method is most needed. Despite this, these issues do not affect the paper’s main contribution: it pinpoints an important offline failure mode that had not been clearly formulated before, provides a principled theoretical analysis, and offers a technically coherent correction with solid empirical support.

Based on the importance of the problem, the clarity of the theoretical contribution, and the overall strength of the empirical results, I recommend **weak accept**.